# Effects of water, sanitation, handwashing, and nutritional interventions on telomere length among children in a cluster-randomized controlled trial in rural Bangladesh

Audrie Lin[1]\*, Benjamin F Arnold[1], Andrew N Mertens[1], Jue Lin[2], Jade Benjamin-Chung[1], Shahjahan Ali[3], Alan E Hubbard[1], Christine P Stewart[4], Abul K Shoab[3], Md Ziaur Rahman[3], Md Saheen Hossen[3], Palash Mutsuddi[3], Syeda L Famida[3], Salma Akther[3], Mahbubur Rahman[3], Leanne Unicomb[3], Firdaus S Dhabhar[5], Lia C H Fernald[1], John M Colford Jnr[1], Stephen P Luby[6]

[1]School of Public Health, University of California, Berkeley, Berkeley, United States; [2]Department of Biochemistry and Biophysics, University of California, San Francisco, San Francisco, United States; [3]Infectious Disease Division, International Centre for Diarrhoeal Disease Research, Dhaka, Bangladesh; [4]Department of Nutrition, University of California, Davis, Davis, United States; [5]Sylvester Comprehensive Cancer Center, Department of Psychiatry and Behavioral Sciences, Miller School of Medicine, University of Miami, Miami, United States; [6]Division of Infectious Diseases and Geographic Medicine, Stanford University, Stanford, United States

**\*For correspondence:**
audrielin@berkeley.edu

## Abstract

**Background:** Shorter childhood telomere length (TL) and more rapid TL attrition are widely regarded as manifestations of stress. However, the potential effects of health interventions on child TL are unknown. We hypothesized that a water, sanitation, handwashing (WSH), and nutritional intervention would slow TL attrition during the first two years of life.

**Methods:** In a trial in rural Bangladesh, we randomized geographical clusters of pregnant women into individual water treatment, sanitation, handwashing, nutrition, combined WSH, combined nutrition plus WSH (N + WSH), or control arms. We conducted a substudy enrolling children from the control arm and the N + WSH intervention arm. Participants and outcome assessors were not masked; analyses were masked. Relative TL was measured at 1 and 2 years after intervention, and the change in relative TL was reported. Analysis was intention-to-treat.

**Results:** Between May 2012 and July 2013, in the overall trial, we randomized 720 geographical clusters of 5551 pregnant women to a control or an intervention arm. In this substudy, after 1 year of intervention, we assessed a total of 662 children (341 intervention and 321 control) and 713 children after 2 years of intervention (383 intervention and 330 control). Children in the intervention arm had significantly shorter relative TL compared with controls after 1 year of intervention (difference −163 base pairs (bp), p=0.001). Between years 1 and 2, TL increased in the intervention arm (+76 bp) and decreased in the controls (−23 bp) (p=0.050). After 2 years, there was no difference between the arms (p=0.305).

**Conclusions:** Our unexpected finding of increased telomere attrition during the first year of life in the intervention group suggests that rapid telomere attrition during this critical period could reflect the improved growth in the intervention group, rather than accumulated stress.

**Funding:** Funded byThe Bill and Melinda Gates Foundation.

**Clinical trial number:** NCT01590095.

## Introduction

Children in low-income countries often experience infectious diseases and nutritional deficiencies leading to impaired growth, poor development, and early mortality (*Black et al., 2017*; *GBD 2015 Mortality and Causes of Death Collaborators, 2016*; *Victora et al., 2008*). During early life, children exhibit heightened developmental plasticity and are more sensitive to environmental conditions than later in life (*Barker, 2007*). The theory of developmental origins of health and disease postulates that multiple, cumulative early life exposures to adverse environmental factors may increase allostatic load (the cumulative biological damage from chronic stress) and susceptibility to adult diseases (*Barker, 2007*; *Juster et al., 2010*; *Price et al., 2013*).

Accumulating evidence implicates telomere length (TL) attrition as a potentially important underlying mechanism that links early life insults with adverse health outcomes later in life (*Price et al., 2013*). Telomeres, the repetitive DNA sequences and protein complexes protecting the ends of linear chromosomes, gradually shorten during normal cell division. Progressive telomere shortening leads to chromosome instability and cell senescence (*Blackburn, 2001*). Shorter TL has been linked to several age-related conditions including diabetes, heart disease, and early mortality (*Cawthon et al., 2003*; *Fitzpatrick et al., 2007*; *Salpea et al., 2010*). It remains an open debate whether TL serves as a 'molecular clock' that gauges cumulative stress exposures over a lifespan or plays a role in the etiology of various diseases (*Blackburn et al., 2015*; *Hamad et al., 2016*; *Zhan et al., 2015*).

Acute or chronic infections may contribute to childhood TL attrition, and inflammation and oxidative stress may be potential mediators (*Houben et al., 2008*). Infections may induce T-cell proliferation and accelerated telomere attrition (*Aviv, 2004*). Although no studies have directly investigated the relationship between infection and childhood TL, related studies in adults and animals support the plausibility of an association. Animal models have demonstrated that repeated exposures to *Salmonella enterica* cause telomere attrition (*Ilmonen et al., 2008*), and prenatal chronic malaria infections shorten offspring TL (*Asghar et al., 2015*). In adult humans, *Helicobacter pylori* infection, hepatitis C virus, HIV, and experimentally induced respiratory infection have been associated with shorter TL (*Cohen et al., 2013*; *Gianesin et al., 2016*; *Hou et al., 2009*; *Zanet et al., 2014*). Furthermore, caregiver-reported diarrhea in the first two years of life predicted shorter adult TL (*Eisenberg et al., 2017*). Early life water, sanitation, and handwashing (WSH) interventions could potentially prevent or reduce infections and slow telomere attrition. A systematic review and meta-analysis of WSH interventions reported a reduction in diarrheal illness (pooled estimate of relative risk 0.67, 95% CI 0.59–0.76) (*Fewtrell et al., 2005*), and a randomized controlled trial in Pakistan showed that promotion of handwashing decreased acute respiratory infections in children (*Luby et al., 2005*). To our knowledge, no studies have examined the impact of WSH interventions on TL.

Early life nutrition may affect childhood TL. Breast milk could potentially reduce telomere attrition by protecting against inflammation and oxidative stress (*Cacho and Lawrence, 2017*; *Matos et al., 2015*) – exposures associated with telomere attrition (*Houben et al., 2008*). Studies have found an association between exclusive breastfeeding and preschool TL (*Wojcicki et al., 2016a*), but no association with adult TL (*Eisenberg et al., 2017*). Improved intake of micronutrients (vitamins and minerals) may promote telomere maintenance (*Bull and Fenech, 2008*). Studies in adult populations have found mixed evidence for associations between TL, multivitamin usage, and various micronutrients (*Cassidy et al., 2010*; *Liu et al., 2013*; *Paul et al., 2015*; *Richards et al., 2007*; *Xu et al., 2009*). To our knowledge, no randomized controlled trials have assessed the effect of nutritional interventions on child TL.

Telomeres shorten fourfold faster in infants compared to adults (*Zeichner et al., 1999*); however, only a few studies have assessed the potential associations between environmental factors and TL in early childhood, a sensitive window of growth and development (*Entringer et al., 2013*; *Marchetto et al., 2016*; *Theall et al., 2013a*; *Theall et al., 2013b*; *Wojcicki et al., 2016a*). The trajectories of infant TL in low-income countries and the potential impact of early life health-improvement interventions on TL are unknown. We conducted a substudy within a randomized trial in rural

**eLife digest** Stress negatively affects health by causing changes in cells. As a result, excess stress may predispose people to fall ill more often or age faster. It is difficult to measure stress. Some studies suggest that measuring the ends of chromosomes, known as telomeres, may be one way to measure stress. Like the plastic tips on shoelaces, telomeres protect chromosomes from fraying. All peoples' telomeres shorten over their lifetime with each cell division. Many studies show that telomeres shorten faster in people who experience more stress. When telomeres become too short, cells die faster without being replaced, and the body ages.

Most studies on telomere length have looked at adults. Few studies have looked at children early in life or asked whether there are ways to intervene to stop or reverse stress-related telomere shortening. The first two years of life are a crucial period for the developing brain and immune system, which could set children on a lifelong course toward health or disease. Young children living in low-resource settings often encounter many sources of stress, like poor nutrition, infectious diseases or violence. Studies are needed to determine if interventions in early childhood aimed at reducing some sources of stress improve telomere length or long-term health.

Now, Lin et al. show that interventions to provide safe water, sanitation, handwashing facilities, and better nutrition to children in rural Bangladesh unexpectedly shortened telomeres. As part of a larger study, pregnant women in rural Bangladesh were divided, at random, into groups. One group received a suite of interventions, which included more sanitary toilets, handwashing facilities, and nutritional supplements for their infants. Another group served as a control and did not receive this extra help. Lin et al. looked at telomere length, growth, and infections in a subset of 713 children whose mothers participated in the study.

Children who got the extra help grew faster and were less likely to get diarrhea or parasitic infections than the children in the control group. Unexpectedly, children in the intervention group had shorter telomeres at 14 months of age than the children in the control group. Lin et al. suggest that the telomere shortening in the intervention group might be a consequence of rapid growth and immune system development in the first year of life rather than resulting from biological stress. More studies are needed to ask whether telomere shortening is indeed linked to faster growth and development early in life. The strong and unexpected findings highlight how little is known about how the length of telomeres can be used to predict future health or disease. Interpreting the length of telomeres over a person's lifetime could prove more nuanced than originally thought.

Bangladesh to evaluate if an intensive, early life nutrition, water, sanitation, and handwashing intervention would slow telomere attrition among children in their first two years of life (*Arnold et al., 2013*).

## Results

The objective of the WASH Benefits trial was to compare the effects of individual and combined interventions on child health in the first two years of life – the critical window to prevent growth faltering (*Arnold et al., 2013*). Between 31 May 2012 and 7 July 2013, in the overall trial, 5551 compounds (collections of related households) with pregnant women in their first or second trimester were randomly allocated to one of the six intervention groups or a double sized control group as follows: (1) chlorinated drinking water and safe storage vessel, (2) upgraded sanitation (child potties, sani-scoop hoes to remove feces, and a double pit latrine with a hygienic water seal), (3) handwashing promotion (handwashing stations with detergent soap), (4) combined water + sanitation + handwashing (5) nutrition (lipid-based nutrient supplements and age-appropriate recommendations on maternal nutrition and infant feeding practices), (6) combined nutrition + water + sanitation + handwashing (N + WSH), (7) control group, which did not receive any interventions (*Figure 1*). Community health promoters visited study compounds in the intervention arms to promote behaviors. We conducted a substudy within the trial to evaluate if the N + WSH intervention would slow telomere attrition among children in their first two years of life (*Arnold et al., 2013*). The substudy team visited 996 index children after 1 year of intervention (Y1) and 1021 children after 2 years of intervention (Y2) in the control and N + WSH arms only. TL outcomes were measured in 66.5% of

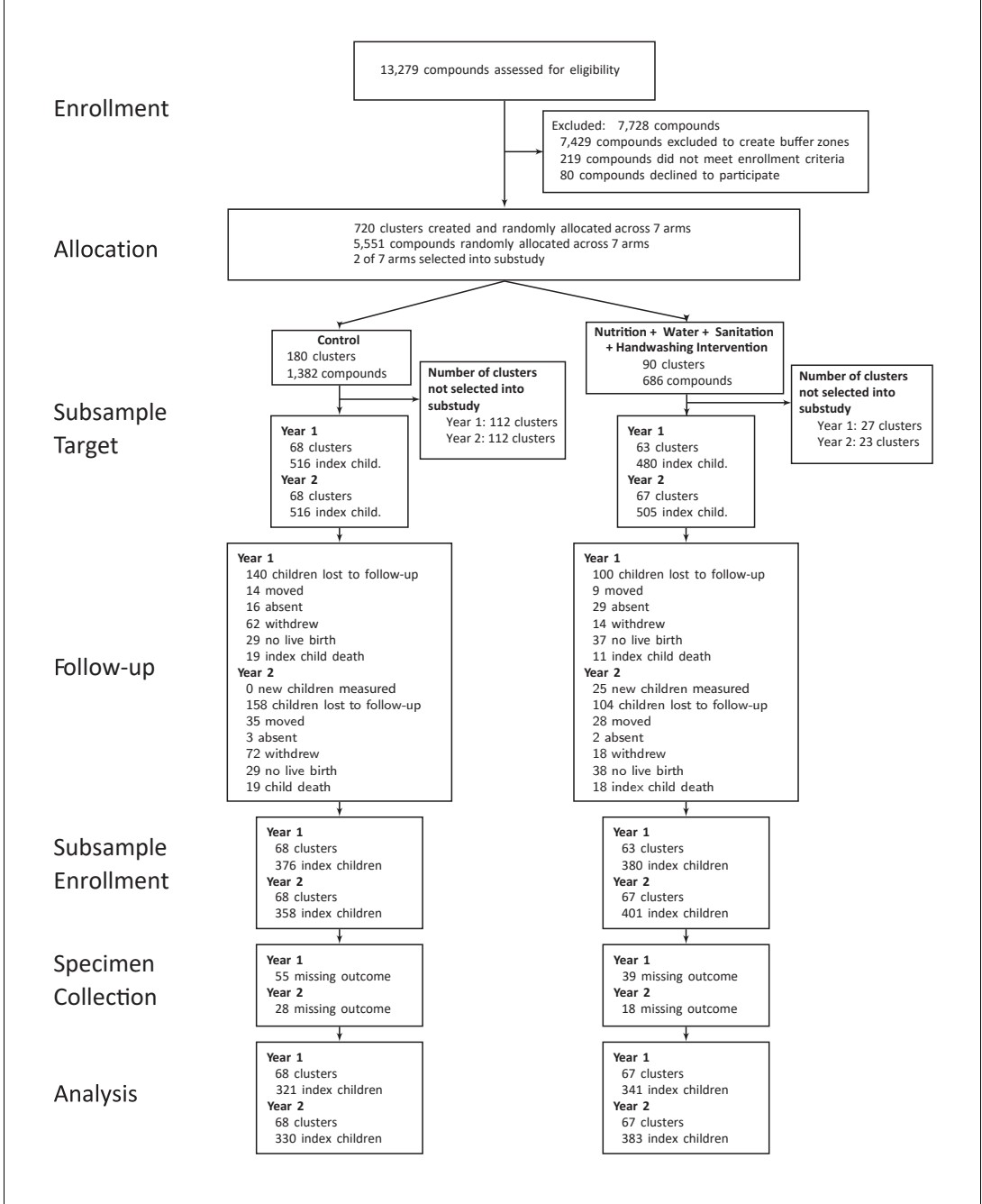

**Figure 1.** Flowchart of participants' progress through the phases of the trial.
The online version of this article includes the following source data for figure 1:

**Source data 1.** Source data and code for *Figure 1*.

the children (N = 662) at Y1% and 69.8% of the children (N = 713) at Y2, but not at birth (*Figure 1*). We expect TL at birth to be similar for the intervention and control arms because household enrollment characteristics were balanced between both arms (*Table 1*).

In the substudy, a quarter of the fathers were engaged in agriculture. 61% of households reported having electricity available, and 86% had an earthen floor. At enrollment, 72% of households were food secure, 56% of households owned a latrine, and 9% of households had a handwashing station with soap near the latrine. The primary water source for the majority of households (72%) was a shallow tubewell. Respondents reported the occurrence of daily open defecation in 80% of

**Table 1.** Enrollment characteristics within the Control households and the N + WSH intervention households

| No. of children: | Children measured at Year 1 | | Children measured at Year 2 | |
|---|---|---|---|---|
| | Control (N=321) | N+WSH Intervention (N=338) | Control (N=330) | N+WSH Intervention (N=380) |
| | % / mean (SD) | % / mean (SD) | % / mean (SD) | % / mean (SD) |
| Maternal | | | | |
| Age (years) | 23 (5) | 24 (5) | 23 (5) | 24 (5) |
| Years of education | 7 (3) | 6 (3) | 7 (3) | 6 (3) |
| Paternal | | | | |
| Years of education | 5 (4) | 5 (4) | 6 (4) | 5 (4) |
| Works in agriculture | 24% | 29% | 25% | 29% |
| Household | | | | |
| Number of persons | 5 (2) | 5 (2) | 5 (2) | 5 (2) |
| Has electricity | 60% | 62% | 62% | 62% |
| Has a cement floor | 16% | 12% | 15% | 13% |
| Acres of agricultural land owned | 0 (0) | 0 (0) | 0 (0) | 0 (0) |
| Drinking Water | | | | |
| Shallow tubewell primary water source | 72% | 70% | 72% | 72% |
| Stored water observed at home | 49% | 53% | 49% | 52% |
| Reported treating water yesterday | 0% | 0% | 0% | 0% |
| Distance (mins) to primary water source | 1 (1) | 1 (2) | 1 (1) | 1 (2) |
| Sanitation | | | | |
| Reported daily open defecation | | | | |
| Adult men | 4% | 9% | 4% | 9% |
| Adult women | 3% | 5% | 3% | 5% |
| Children: 8-<15 years | 4% | 11% | 2% | 11% |
| Children: 3-<8 years | 29% | 35% | 32% | 37% |
| Children: 0-<3 years | 73% | 88% | 73% | 88% |
| Latrine | | | | |
| Owned | 62% | 52% | 59% | 52% |
| Concrete slab | 97% | 93% | 96% | 94% |
| Functional water seal | 38% | 31% | 38% | 31% |
| Visible stool on slab or floor | 54% | 48% | 52% | 46% |
| Owned a potty | 8% | 4% | 7% | 5% |
| Human feces observed in the | | | | |
| House | 6% | 7% | 5% | 9% |
| Child's play area | 1% | 1% | 1% | 2% |
| Handwashing | | | | |
| Within 6 steps of latrine | | | | |
| Has water | 18% | 13% | 21% | 14% |
| Has soap | 9% | 6% | 11% | 7% |
| Within 6 steps of kitchen | | | | |
| Has water | 11% | 10% | 11% | 11% |
| Has soap | 3% | 3% | 5% | 4% |
| Nutrition | | | | |
| *Household is food secure | 74% | 72% | 73% | 72% |

Enrollment characteristics of households with children who had telomere measurements. Data are percentages of binary variables or mean (SD) of continuous variables. Percentages were estimated from slightly smaller denominators than those shown at the top of the table for the following variables due to missing values: mother's age, father's education, father works in agriculture, acres of land owned, open defecation, latrine has a concrete slab, latrine has a functional water seal, visible stool on latrine slab or floor, ownership of child potty, observed feces in the house or child's play area, handwashing variables.
*Assessed by the Household Food Insecurity Access Scale

The online version of this article includes the following source data for Table 1:
Source data 1. Source data and code for *Table 1*.

children less than 3 years of age. The substudy household enrollment characteristics were similar to the overall trial (*Table 2*).

We measured whole blood relative telomere length by quantitative polymerase chain reaction (qPCR), expressed as the ratio of telomere to single-copy gene abundance (T/S ratio) (*Cawthon, 2002*; *Lin et al., 2010*). At the time of TL assessment, the mean (±SD) age was 14.1 (±2.1) months at Y1 and 28.2 (±1.9) months at Y2. TL was normally distributed (*Figure 2*). The mean (±SD) TL was 1.43 (±0.23) T/S ratio at Y1 (6729 ± 549 base pairs (bp)) and 1.45 (±0.24) T/S ratio at Y2 (6763 ± 586 bp). Although these averages are short for young children compared to some previous findings (*Factor-Litvak et al., 2016*; *Frenck et al., 1998*; *Wojcicki et al., 2016a*), they are similar to a recent report of newborns whose mothers experienced a high level of stress during pregnancy (*Marchetto et al., 2016*); the stress associated with low socioeconomic status may have contributed to the short average TL in this study.

We conducted an intention-to-treat analysis using generalized linear models with robust standard errors. Measures of intervention adherence (including observed hardware availability) were high (over 80%) and sustained throughout Y1 and Y2 (S. P. Luby *et al.* in review). After 1 year of intervention, compared with controls (mean = 1.47 T/S or 6813 bp), children in the combined intervention arm (mean = 1.40 T/S or 6650 bp) had significantly shorter relative TL (difference −0.07 T/S or −163 bp, p=0.001; *Table 3*). The entire distribution of the T/S ratio was shifted lower for the intervention arm compared to the control at Y1 (*Figure 2*). Unadjusted, adjusted, and inverse probability of censoring weighting (IPCW; to adjust for potential bias from loss to follow-up) analyses yielded similar estimates (*Table 3*). At the Y2 measurement, there was no significant difference (p=0.305) between the intervention arm and the control (*Table 3*, *Figure 2*).

Next, we compared the change in TL between Y1 and Y2. Household enrollment characteristics were balanced between individuals who had TL outcomes at Y1 versus those who were lost to follow-up at Y2 (*Table 2*). Of index children with a TL measurement at both Y1 and Y2 (N = 557; 54.6% of the total children visited), relative TL increased by +0.03 T/S (+76 bp) in the intervention group and decreased by −0.01 T/S (−23 bp) in the control group (*Table 3*). Overall, the difference between the control and intervention arms in the change in relative TL from Y1 to Y2 was 0.04 T/S and was borderline significant (p=0.050; *Table 3*). Unadjusted, adjusted, and inverse probability of censoring weighting analyses generated similar estimates but only the unadjusted model was significant (*Table 3*).

Finally, we conducted a subgroup analysis by sex because biological differences, differential care practices by sex, or other sex-specific behaviors could modify the effect of the intervention on TL when stratifying by sex (*Wojcicki et al., 2016b*). Sex was not a significant effect modifier (sex by treatment interaction p=0.435 at Y1 and p=0.105 at Y2; *Table 4*). Boys had shorter TL than girls at Y1 (−0.07 T/S, p=0.007) and Y2 (−0.09 T/S, p=0.001), consistent with previous studies (*Gardner et al., 2014*). Although a previous study reported seasonal variation in TL (*Rehkopf et al., 2014*), we did not observe seasonal changes within this study.

## Discussion

Here, we demonstrate an effect of an intervention on TL in infants and report the trajectories of infant TL in a low-income country. In rural Bangladesh, an intensive, combined water, sanitation, handwashing, and nutrition intervention delivered to compounds of newborn children increased TL attrition during the critical first year of life. This result is contrary to many studies that reported increased TL attrition associated with prenatal psychosocial stress (*Entringer et al., 2013*; *Marchetto et al., 2016*), childhood institutional care (*Drury et al., 2012*), disease (*Fitzpatrick et al.,*

**Table 2.** Balance of enrollment characteristics in the WASH Benefits main trial, telomere substudy children enrolled at Year 1, and telomere substudy children lost to follow-up at Year 2

| No. of children: | WASH Benefits Main Trial | | Telomere substudy: Had telomere outcomes at Year 1 | | Telomere substudy: Lost to follow-up at Year 2 (from those who had telomere outcomes at Year 1) | |
|---|---|---|---|---|---|---|
| | Control (N=1779) | N+WSH Intervention (N=953) | Control (N=321) | Control (N=338) | N+WSH Intervention (N=61) | N+WSH Intervention (N=44) |
| | % / mean (SD) | % / mean (SD) | % / mean (SD) | % / mean (SD) | % / mean (SD) | % / mean (SD) |
| Maternal | | | | | | |
| Age (years) | 24 (5) | 24 (6) | 23 (5) | 24 (5) | 23 (4) | 23 (5) |
| Years of education | 6 (3) | 6 (3) | 7 (3) | 6 (3) | 7 (3) | 6 (4) |
| Paternal | | | | | | |
| Years of education | 5 (4) | 5 (4) | 5 (4) | 5 (4) | 5 (4) | 5 (4) |
| Works in agriculture | 30% | 30% | 24% | 29% | 20% | 18% |
| Household | | | | | | |
| Number of persons | 5 (2) | 5 (2) | 5 (2) | 5 (2) | 5 (3) | 5 (2) |
| Has electricity | 57% | 60% | 60% | 62% | 57% | 61% |
| Has a cement floor | 10% | 10% | 16% | 12% | 20% | 7% |
| Acres of agricultural land owned | 0.15 (0.21) | 0.14 (0.38) | 0 (0) | 0 (0) | 0 (0) | 0 (0) |
| Drinking Water | | | | | | |
| Shallow tubewell primary water source | 75% | 73% | 72% | 70% | 77% | 66% |
| Stored water observed at home | 48% | 48% | 49% | 53% | 56% | 57% |
| Reported treating water yesterday | 0% | 0% | 0% | 0% | 0% | 0% |
| Distance (mins) to primary water source | 1 (1) | 1 (2) | 1 (1) | 1 (2) | 1 (1) | 1 (1) |
| Sanitation | | | | | | |
| Reported daily open defecation | | | | | | |
| Adult men | 7% | 7% | 4% | 9% | 3% | 7% |
| Adult women | 4% | 4% | 3% | 5% | 2% | 2% |
| Children: 8-<15 years | 10% | 10% | 4% | 11% | 8% | 13% |
| Children: 3-<8 years | 38% | 37% | 29% | 35% | 30% | 32% |
| Children: 0-<3 years | 82% | 88% | 73% | 88% | 73% | 83% |
| Latrine | | | | | | |
| Owned | 54% | 53% | 62% | 52% | 67% | 50% |
| Concrete slab | 95% | 94% | 97% | 93% | 100% | 98% |
| Functional water seal | 31% | 27% | 38% | 31% | 46% | 38% |
| Visible stool on slab or floor | 48% | 46% | 54% | 48% | 64% | 53% |
| Owned a potty | 4% | 4% | 8% | 4% | 16% | 2% |
| Human feces observed in the | | | | | | |
| House | 8% | 7% | 6% | 7% | 10% | 5% |
| Child's play area | 2% | 1% | 1% | 1% | 2% | 0% |
| Handwashing | | | | | | |
| Within 6 steps of latrine | | | | | | |

*Table 2 continued on next page*

*Table 2 continued*

| No. of children: | WASH Benefits Main Trial | | Telomere substudy: Had telomere outcomes at Year 1 | | Telomere substudy: Lost to follow-up at Year 2 (from those who had telomere outcomes at Year 1) | |
| --- | --- | --- | --- | --- | --- | --- |
| | Control (N=1779) | N+WSH Intervention (N=953) | Control (N=321) | Control (N=338) | N+WSH Intervention (N=61) | N+WSH Intervention (N=44) |
| | % / mean (SD) | % / mean (SD) | % / mean (SD) | % / mean (SD) | % / mean (SD) | % / mean (SD) |
| Has water | 14% | 11% | 18% | 13% | 18% | 8% |
| Has soap | 7% | 6% | 9% | 6% | 11% | 8% |
| Within 6 steps of kitchen | | | | | | |
| Has water | 9% | 9% | 11% | 10% | 9% | 5% |
| Has soap | 3% | 3% | 3% | 3% | 0% | 0% |
| Nutrition | | | | | | |
| *Household is food secure | 67% | 71% | 74% | 72% | 75% | 68% |

Data are percentages of binary variables or mean (SD) of continuous variables. Percentages were estimated from slightly smaller denominators than those shown at the top of the table for the following variables due to missing values: mother's age, father's education, father works in agriculture, acres of land owned, open defecation, latrine has a concrete slab, latrine has a functional water seal, visible stool on latrine slab or floor, ownership of child potty, observed feces in the house or child's play area, handwashing variables.

*Assessed by the Household Food Insecurity Access Scale

The online version of this article includes the following source data for Table 2:

**Source data 1.** Source data and code for *Table 2*.

2007; *Salpea et al., 2010*), and mortality (*Cawthon et al., 2003*); however, these prior studies did not examine participants in the same age range (1–2 years) as this study. The difference between the intervention and control arms in the change in relative TL from Y1 to Y2 was borderline significant, and at Y2, we found no difference in relative TL between the arms. In high- and low-resource settings, there is a dearth of evidence on the effect of environmental exposures on TL trajectories in this age range and its implications for adult health outcomes.

The accelerated TL attrition observed during the first year of life in the intervention arm may reflect improved immune system development. The WASH Benefits trial found that children in the intervention arm had reductions in caregiver-reported diarrhea, soil-transmitted helminth infections, and *Giardia duodenalis* infections compared with the control arm (S. P. Luby *et al.*, A. Ercumen *et al.*, and A. Lin *et al.* in review). It is plausible that the interventions improved immune system development resulting in subsequent protection against infections or that the decreased exposure to pathogens may have improved immunity. The concept of accelerated TL attrition during the first year of life as a reflection of better immune system development is consistent with studies that have reported rapid telomere attrition and accelerated differentiation of hematopoietic stem cells (HSCs), stem cells that give rise to all lineages of immune cells, during the first year of life. A healthy individual experiences ~ 17 HSC divisions in the first year of life, followed by ~2.5 divisions/year between ages 3–13 years, and ~0.6 divisions/year in adults (*Elwood, 2004*; *Kimura et al., 2010*; *Rufer et al., 1999*; *Sidorov et al., 2009*). Since TL shortens in proportion to the number of cell replications, HSC progenitor divisions correspond with TL attrition rates (*Elwood, 2004*; *Kimura et al., 2010*; *Rufer et al., 1999*; *Sidorov et al., 2009*). In the first year of life, children in the control group with poor immune system development would potentially experience less HSC divisions and slower rates of TL attrition compared to the intervention group.

The accelerated TL attrition during the first year of life in the intervention arm may also reflect better linear growth. The intervention improved child linear growth at both Y1 and Y2 measurements (S. P. Luby *et al.* in review). An evolutionary adaptation of organisms during infections is to restrict growth, and instead redirect nutrients and energy to ensure survival (*O'Connor et al., 2008*). We postulate that the decreased pathogen exposure and the subsequent reduction in acquired infections are potential mechanisms by which better growth was attained in the intervention arm. Some studies have reported high synchrony between peripheral blood and TL attrition rates within other somatic tissues in the same individual (*Daniali et al., 2013*; *Takubo et al., 2002*), while others have

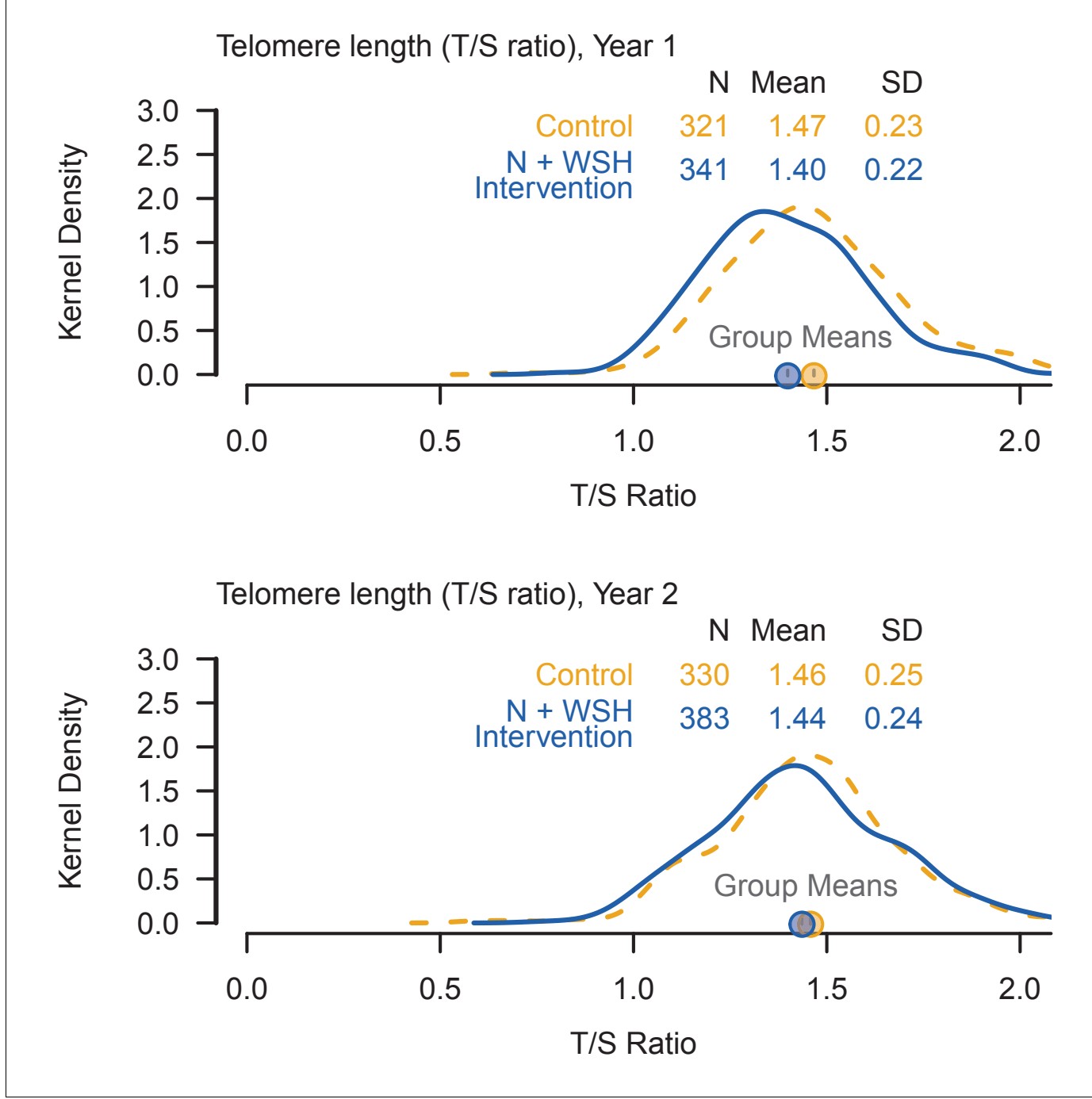

**Figure 2.** Kernel density plots summarize the distribution of the telomere lengths (T/S ratios) among enrolled children. In each panel, a dashed orange line illustrates the distribution of T/S ratio among control children and a solid blue line illustrates the distribution of T/S ratio among N+WSH intervention children. Even if a child was not present at Year 1, they were included in the analysis if they provided a sample at Year 2.

The online version of this article includes the following source data for figure 2:

**Source data 1.** Source data and code for *Figure 2*.

found differences (*Dlouha et al., 2014*; *Thomas et al., 2008*). Although the results of these prior investigations are equivocal, the accelerated peripheral blood TL attrition in the intervention group during the first year of life could potentially indicate rapid muscle or bone cell division involved in growth. We hypothesize that early life TL may be a proxy measure for growth and development of the immune system, the brain, and other vital tissues. Forthcoming WASH Benefits studies focused

**Table 3.** Effect of intervention on telomere length (T/S ratio) after 1 and 2 years of N + WSH intervention and on change in telomere length between Year 1 and Year 2.

| Arm | N | Mean | Unadjusted difference: Intervention vs. Control (95% CI) | Age- and sex- adjusted difference: Intervention vs. Control (95% CI) | Fully adjusted [†] difference: Intervention vs. Control (95% CI) | Inverse probability of censoring [†] difference: Intervention vs. Control (95% CI) |
|---|---|---|---|---|---|---|
| After 1 year of intervention (age ~ 14 months) | | | | | | |
| Control | 321 | 1.47 | | | | |
| N + WSH | 341 | 1.40 | −0.07 (−0.11,−0.03) p=0.001* | −0.06 (−0.10,−0.02) p=0.005* | −0.05 (−0.10,0.01) p=0.099 | −0.07 (−0.11,−0.03) p=0.001* |
| After 2 years of intervention (age ~ 28 months) | | | | | | |
| Control | 330 | 1.46 | | | | |
| N + WSH | 383 | 1.44 | −0.02 (−0.06,0.02) p=0.305 | −0.02 (−0.06,0.02) p=0.271 | −0.04 (−0.09,0.01) p=0.147 | −0.02 (−0.06,0.02) p=0.288 |
| Change in telomere length between Year 1 and 2 | | | | | | |
| Control | 260 | −0.01 | | | | |
| N + WSH | 297 | 0.03 | 0.04 (0.00,0.08) p=0.050* | 0.04 (−0.00,0.08) p=0.073 | 0.04 (−0.01,0.09) p=0.081 | 0.03 (−0.01,0.07) p=0.088 |

*P<0.05

Confidence intervals were adjusted for clustered observations using robust standard errors. Separate weights were created for the inverse probability weighting for each of the different analyses because the probability of missing at Year 1 was different than at Year 2.

† Adjusted for pre-specified covariates associated with the outcome (likelihood ratio test p-value<0.2): Field staff who collected data, month of measurement, household food insecurity, child age, child sex, mother's age, mother's height, mother's education level, number of children < 18 years in the household, number of individuals living in the compound, distance in minutes to the primary water source, household floor materials, household wall materials, household electricity, and household assets (wardrobe, table, chair, clock, khat, chouki, radio, television, refrigerator, bicycle, motorcycle, sewing machine, mobile phone, cattle, goats, and chickens).

The online version of this article includes the following source data for Table 3:

Source data 1. Source data and code for *Table 3*.

on child inflammation and enteric pathogen burden will provide evidence on mechanistic pathways leading to different TL, growth, and development outcomes.

**Table 4.** Subgroup analysis across sex of the effect of N + WSH intervention on telomere length (T/S ratio) after 1 and 2 years of intervention and on change in telomere length between Year 1 and Year 2

| Arm | Female children | | Male children | | Female children Unadjusted difference: Intervention vs. Control (95% CI) | Male children Unadjusted difference: Intervention vs. Control (95% CI) | Interaction term sex by treatment (95% CI) |
|---|---|---|---|---|---|---|---|
| | N | Mean | N | Mean | | | |
| After 1 year of intervention (age ~ 14 months) | | | | | | | |
| Control | 162 | 1.50 | 159 | 1.43 | | | |
| N + WSH | 180 | 1.42 | 161 | 1.37 | −0.08 (−0.13,−0.03) p=0.001* | −0.06 (−0.11,−0.00) p=0.040* | 0.03 (−0.04,0.09) p=0.435 |
| After 2 years of intervention (age ~ 28 months) | | | | | | | |
| Control | 167 | 1.50 | 163 | 1.42 | | | |
| N + WSH | 202 | 1.45 | 181 | 1.42 | −0.05 (−0.10,0.01) p=0.082 | 0.00 (−0.05,0.05) p=0.877 | 0.05 (−0.01,0.12) p=0.105 |
| Change in telomere length between Year 1 and 2 | | | | | | | |
| Control | 134 | −0.01 | 126 | −0.01 | | | |
| N + WSH | 160 | 0.03 | 137 | 0.04 | 0.04 (−0.02,0.09) p=0.227 | 0.05 (0.00,0.09) p=0.048* | 0.01 (−0.06,0.08) p=0.747 |

*P<0.05

Confidence intervals were adjusted for clustered observations using robust standard errors.

The online version of this article includes the following source data for Table 4:

Source data 1. Source data and code for *Table 4*.

After the initial period of rapid linear growth during Y1, our borderline significant finding that telomere length increased in the intervention arm and decreased in the control arm from Y1 to Y2 seems to represent a preview of the TL trajectories that are set: we hypothesize that the intervention children will experience slower lifetime TL attrition and more physiological resistance to stress-related diseases (*Juster et al., 2010*). The change in relative TL from Y1 to Y2 was small (0.04 T/S 95% confidence interval 0.00, 0.08) and the p-value borderline (p=0.050). Nevertheless, our diarrhea, soil-transmitted helminth, *Giardia duodenalis*, and growth results suggest that these interventions are potentially interrupting key infection and malnutrition pathways to reduce exposure to biological adversity and allostatic load (S. P. Luby *et al.*, A. Ercumen *et al.*, and A. Lin *et al.* in review). In a Filipino cohort, decreased infant diarrheal prevalence was associated with longer adult TL (*Eisenberg et al., 2017*). In observational studies, various vitamins were associated with longer adult TL (*Xu et al., 2009*), and early exclusive breastfeeding was associated with longer child TL at ages 4–5 years (*Wojcicki et al., 2016a*). Our nutrition intervention included promotion of exclusive breastfeeding and micronutrient fortified lipid-based nutrient supplements. The Y1 to Y2 changes in TL that we observed are consistent with these studies. The smaller than expected treatment effect and the borderline p-value indicate that the trial was slightly underpowered to detect differences in the change in relative TL from Y1 to Y2 among the intervention and control groups.

Alternatively, our finding of a modest, borderline impact from Y1 to Y2 may suggest that the effect of the interventions is diminished in children over 1 year of age. This potential waning of intervention effects as a child ages could be due to less consumption of breast milk and thus, reductions in its protective effects or the increased mobility of children in environments contaminated with animal feces. This interpretation would also be consistent with no differences in relative TL observed at Y2 between the two groups, underscoring the importance of targeting interventions early in life during the sensitive period when they are likely to have the largest impact on childhood TL.

This study had several limitations. These results from a rural, low-resource area in Bangladesh may not generalize to other populations. The lack of geographic matching in the substudy could have resulted in imbalances between study arms in factors associated with geography. In addition, differential loss to follow-up in the control and intervention arms may have biased results due to differences in unmeasured characteristics. However, our IPCW analysis showed that bias from loss to follow-up was unlikely based on a large set of measurable characteristics. The lithium heparin additive in the whole blood may have inhibited the qPCR reaction and the long duration of time from sample collection to TL measurement may have affected TL, but any systematic measurement errors would likely affect both study groups equally. These factors may provide potential explanations for the shorter average TL observed in this study compared to other studies (*Factor-Litvak et al., 2016*; *Frenck et al., 1998*; *Wojcicki et al., 2016a*). We only measured TL in whole blood, which might yield different results compared to TL measured in less proliferative tissue types (e.g., muscle or fat) (*Daniali et al., 2013*). Finally, we did not measure TL at birth (*Wojcicki et al., 2016b*), but we would expect it to be similar for both groups because household characteristics were balanced by randomization.

Our findings are surprising, and they represent important contributions to the nascent and rapidly expanding field of infant telomere biology. Most studies to date have highlighted shorter TL as a measure of intrauterine or childhood stress (*Drury et al., 2012*; *Entringer et al., 2013*; *Marchetto et al., 2016*; *Price et al., 2013*), disease (*Fitzpatrick et al., 2007*; *Salpea et al., 2010*), and mortality (*Cawthon et al., 2003*). However, our results motivate an intriguing hypothesis: here, we suggest that during the first year of life, accelerated TL attrition could reflect better child growth, neurodevelopment, and immune function. Although TL was a sensitive outcome that responded to an intervention, this trial underscores our limited understanding of environmental exposures contributing to TL dynamics during early life and the critical physiological pathways linking TL and lifelong health trajectories. Since potential confounding could plague observational studies, evaluating the relationship between modifiable exposures and TL within the context of randomized controlled trials could provide valuable contributions to the field of telomere biology.

## Materials and methods

### Study design

We conducted a cluster-randomized trial in the rural subdistricts in Gazipur, Mymensingh, Tangail and Kishoreganj districts of Bangladesh. The main trial enrolled geographically matched clusters of compounds that were randomly allocated to a double-sized control or one of the six intervention arms (Arnold et al., 2013). Each compound in rural Bangladesh consists of a collection of households of extended families. Due to logistical constraints for specimen collection, the substudy enrolled a subsample of randomized clusters that was balanced across arms (allocation ratio 1:1), but not geographically matched.

### Participants

We enrolled pregnant mothers in their first or second trimester. Exclusion criteria included households with plans to move in the following year, households that did not own their home, and households that drew water from a source with high iron content. The children born to the enrolled pregnant mothers were considered index children and are the focus of this analysis. In this substudy, if any two of the following criteria for moderate to severe dehydration were met, the child was excluded from the venipuncture: (1) restless, irritable, (2) sunken eyes, (3) drinks eagerly, thirsty, (4) pinched skin returns to normal position slowly. Children were also excluded from the venipuncture if they were listless or unable to perform their normal activities. Two children met the exclusion criteria: one child in the control arm was excluded at Year 1, and one child in the intervention arm was excluded at Year 2.

### Ethics

Primary caregivers of all children provided written informed consent. The study protocols were approved by human subjects committees at icddr,b (PR-11063 and PR-14108), the University of California, Berkeley (2011-09-3652 and 2014-07-6561) and Stanford University (25863 and 35583). A data safety monitoring committee convened by icddr,b oversaw the study.

### Randomization and masking

We formed clusters of 8 neighboring households with eligible pregnant women and created a 1 km buffer around each cluster to prevent spillover between clusters. A block was the equivalent of eight geographically-adjacent clusters. An investigator at UC Berkeley (B.F.A.) used a random number generator to block randomize clusters to one of the six interventions or to a double sized control arm as follows: (1) drinking water treatment and safe storage, (2) sanitation, (3) handwashing, (4) combined water + sanitation + handwashing (WSH) (5) nutrition, (6) combined nutrition + water + sanitation + handwashing (N + WSH) and (7) non-intervention control group. This substudy only included children in the control and the combined N + WSH arms.

Participants and the data collection team were not masked because each intervention delivered had visible hardware. One laboratory investigator (J.L.), who was masked to group assignments, conducted all of the TL measurements. Two investigators (A.L., A.N.M.) conducted independent masked statistical analyses to generate final estimates following the pre-registered analysis protocol. After all masked analyses were replicated, the results were unmasked.

### Procedures

The combined N + WSH interventions were previously described (Arnold et al., 2013). Briefly, the components of the combined intervention were as follows: water treatment (Aquatabs; NaDCC) and safe storage vessel, sanitation (child potties, sani-scoop hoes to remove feces, and a double pit latrine with a hygienic water seal), handwashing (handwashing stations near the latrine and kitchen, including soapy water bottles and detergent soap), and nutrition (lipid-based nutrient supplements [Nutriset, Malauny, France] that included ≥100% of the recommended daily allowance of 12 vitamins and 9 minerals with 9.6 g of fat and 2.6 g of protein daily for children 6 to 24 months of age and age-appropriate recommendations on maternal nutrition and infant feeding practices). Community health promoters visited study compounds in the intervention arms at least once per week during

the first 6 months and at least once biweekly to promote behaviors. The control group received no intervention.

TL was measured at 1 year and 2 years after intervention delivery when the children were approximately ages 14 and 28 months respectively. For each child, trained icddr,b staff collected a 5 ml venipuncture sample from the antecubital area of the arm into a Sarstedt S-monovette lithium heparin collection tube. All specimens were labeled with identification numbers only. Specimens were transported on ice to the laboratory, immediately centrifuged, and whole blood was stored at −80° C. Specimens were shipped on dry ice (−79°C) to Dr. Elizabeth Blackburn's laboratory at the University of California, San Francisco. The duration of time from sample collection to TL measurement ranged from 8 to 32 months.

## Measurement of relative TL

Genomic DNA was extracted from heparin-anti-coagulated whole blood stored at −80°C using QIAamp DNA Mini Kit (QIAGEN, Hilden, Germany). DNA was quantified by measuring OD260 with a NanoDrop 200 c Spectrophotometer (Nanodrop Products, Wilmington, DE). Samples that passed the quality control of OD260/OD280 between 1.7–2.0 were used for TL measurement. Of the 1384 samples, eight did not pass quality control. Of the remaining 1376 samples assayed, one sample was not amplified, resulting in 1375 samples with valid TL data.

TL was measured in whole blood by quantitative PCR (qPCR) using a protocol adapted from the published original method by Cawthon (*Cawthon, 2002*; *Lin et al., 2010*). This method determines relative TL (i.e., T/S ratio) by measuring the factor by which each DNA sample differs from a reference DNA sample in its ratio of telomere repeat copy number (T) to single-copy gene copy number (S) (*Cawthon, 2002*). The primers for the telomere PCR were *tel1b* [5'-CGGTTT(GTTTGG)$_5$GTT-3'], used at a final concentration of 100 nM, and *tel2b* [5'-GGCTTG(CCTTAC)$_5$CCT-3'], used at a final concentration of 900 nM. The primers for the single-copy gene (human beta-globin) PCR were *hbg1* [5' GCTTCTGACACAACTGTGTTCACTAGC-3'], used at a final concentration of 300 nM, and *hbg2* [5'-CACCAACTTCATCCACGTTCACC-3'], used at a final concentration of 700 nM. The final reaction mix contained 20 mM Tris-HCl, pH 8.4; 50 mM KCl; 200 μM each dNTP; 1% DMSO; 0.4x Syber Green I; 22 ng E. coli DNA; 0.4 Units of Platinum Taq DNA polymerase (Invitrogen Inc., Carlsbad, CA), and approximately 6.6 ng of genomic DNA per 11 microliter reaction.

The telomere (T) thermal cycling qPCR profile consisted of denaturing at 96°C for 1 min followed by 30 cycles of denaturing at 96°C for 1 s and annealing or extension at 54°C for 60 s with fluorescence data collection. The single-copy gene (S) thermal cycling qPCR profile consisted of denaturing at 96°C for 1 min followed by 8 cycles of denaturing at 95°C for 15 s, annealing at 58°C for 1 s, and extension at 72°C for 20 s, followed by 35 cycles of denaturing at 96°C for 1 s, annealing at 58°C for 1 s, extension at 72°C for 20 s, and holding at 83°C for 5 s with data collection. The T/S ratio for each sample was measured twice (technical replicates). When the duplicate T/S value and the initial value varied by more than 7%, the sample was run a third time and the two closest values were reported. The average coefficient of variation for TL measurement in this study was 2.1%.

Tubes containing 26, 8.75, 2.9, 0.97, 0.324 and 0.108 ng of a reference DNA (pooled genomic DNA from 100 females) were included in each PCR run so that the quantity of targeted templates in each research sample could be determined relative to the reference DNA sample by the standard curve method. The same reference DNA was used for all PCR runs. To control for inter-assay variability, eight control DNA samples were included in each run. In each batch, the T/S ratio of each control DNA was divided by the average T/S for the same DNA from 10 runs to calculate a normalizing factor. This is done for all eight samples and the average normalizing factor for all eight samples was used to correct the participant DNA samples to calculate the final T/S ratio. The assay was performed as plates of 96 samples. Control and intervention samples were randomly interspersed to minimize potential plate effects. DNA extraction and TL assay were performed in two batches (3.5 months apart). The first batch contained 663 samples and the second batch contained 721 samples. All assays were performed using the same lots of reagents. To adjust for assay batch variations, 48 samples from the first batch were re-assayed together with the second batch of samples and data from the second batch of samples were adjusted based on the systematic difference between the first batch values versus the second batch values for these 48 samples. This adjustment factor was 1.05.

To determine the conversion factor for the calculation of approximate base pair telomere length from T/S ratio, the above method was used to determine the T/S ratios, relative to the same reference DNA, for a set of genomic DNA samples from the ATCC authenticated human fibroblast primary cell line IMR-90 (ATCC: CCL-186, cell line authentication method: STR profiling; ATCC determined the cell line was free of mycoplasma contamination) at different population doublings, as well as with the telomerase protein subunit gene (hTERT) transfected into a lentiviral construct. The mean TRF length from these DNA samples was determined using Southern blot analysis. Comparison of T/S ratios versus base pairs derived from the Southern blot analysis generated the following equation for conversion from T/S ratio to base pairs: base pairs = 3274 + 2413 * (T/S) (*Farzaneh-Far et al., 2010*).

## Outcomes

The pre-specified outcome measures were TL (T/S ratios) measured at 1 year and 2 years after intervention delivery (Y1 and Y2 respectively), and the change in TL from Y1 to Y2.

## Statistical analysis

The complete pre-registered analysis protocol is available (https://osf.io/cjjwa/). We provide a summary of our analyses below. Analyses were conducted using R statistical software version 3.2.3 (www.r-project.org).

### Sample size

Assuming a standard deviation of 887 base pairs (*Entringer et al., 2013*), a range of cluster-level intra-class correlations for repeated measures (0.01 to 0.20), and a two-sided alpha of 5%, the substudy had 90% power to detect a 207 to 273 base pair difference between the N + WSH arm and the control.

### Statistical parameters

All analyses were intention-to-treat. We compared the N + WSH arm versus the control separately at each time point: 1 year after intervention initiation (Y1; median age 14 months) and 2 years after intervention initiation (Y2; median age 28 months). Between the N + WSH versus control arms, we also compared the change in TL between Y1 and Y2.

The pre-registered analytic approach for these analyses generally followed the same methods as described for the main trial outcomes (*Arnold et al., 2013*). We used generalized linear models with robust standard errors that account for repeated measures within clusters and reported two-tailed p-values. Randomization led to balance in observed covariates across arms, so, in accordance with our pre-specified analysis plan, we relied on the unadjusted analysis as our primary analysis for the TL outcomes. We conducted two sets of secondary adjusted analyses: (1) adjusted for child age and sex only and (2) fully adjusted for child age, sex, and significantly related covariates (likelihood ratio test p-value<0.20).

We conducted a pre-specified subgroup analysis stratified by sex because biological differences, differential care practices, or other behavioral practices could modify the effect of the combined N + WSH intervention when stratifying by gender.

### Missing outcomes

The study carefully tracked enrolled participants and the recovery rates of whole blood specimens per participant. We compared rates of missing blood specimens across randomized arms and also the characteristics of those with missing specimens versus those with a full set of specimens to determine whether missing specimen rates were random. Using the full data (from the main trial) of enrolled mothers in the double-sized control and N + WSH arms, as if every index child with a live birth had TL measured, we used inverse probability weighting to correct for potential bias due to informative censoring (*Little et al., 2012*). The inverse probability of censoring weighted (IPCW) approach follows best practices for removing potential bias owing to missing outcome measurements in trials, according to a definitive methodologic review conducted by the National Research Council (*Little et al., 2012*). Logistic regression models were used to predict the probability of TL measurement given baseline characteristics. The baseline characteristics used to fit the models were

selected by the same covariate prescreening method used in the fully adjusted analysis. TL differences between the control and intervention group were then estimated using adjusted linear regression weighted by the inverse probability of TL measurement.

### Registration
The trial was registered at ClinicalTrials.gov (NCT01590095).

### Role of the funding source
The funder approved the study design, but was not involved in data collection, analysis, interpretation or any decisions related to publication. The corresponding author had full access to all study data and final responsibility for the decision to submit for publication.

### Data availability
The WASH Benefits data and code that support the findings of this study are available in Open Science Framework (https://osf.io/evc98/).

## Acknowledgements

We are indebted to the families who participated in the study and the incredible dedication of the icddr,b staff who delivered the interventions and collected the data and specimens. This study was funded by Global Development grant OPPGD759 from the Bill and Melinda Gates Foundation to the University of California, Berkeley. icddr,b is grateful to the Governments of Bangladesh, Canada, Sweden, and the United Kingdom for providing core/unrestricted support.

## Additional information

### Competing interests
Jue Lin: Co-founder of Telomere Diagnostics Inc., a telomere measurement company. The other authors declare that no competing interests exist.

### Funding

| Funder | Grant reference number | Author |
| --- | --- | --- |
| Bill and Melinda Gates Foundation | Global Development Grant (OPPGD759) | John M Colford Jr |

The funder approved the study design, but was not involved in data collection, analysis, interpretation or any decisions related to publication. The corresponding author had full access to all study data and final responsibility for the decision to submit for publication.

### Author contributions
Audrie Lin, Conceptualization, Data curation, Formal analysis, Funding acquisition, Validation, Investigation, Methodology, Writing—original draft, Project administration, Writing—review and editing; Benjamin F Arnold, Resources, Data curation, Software, Formal analysis, Supervision, Funding acquisition, Validation, Investigation, Methodology, Project administration, Writing—review and editing; Andrew N Mertens, Resources, Data curation, Software, Formal analysis, Validation, Visualization, Methodology, Writing—review and editing; Jue Lin, Resources, Data curation, Validation, Investigation, Methodology, Writing—review and editing; Jade Benjamin-Chung, Resources, Data curation, Software, Funding acquisition, Validation, Investigation, Project administration, Writing—review and editing; Shahjahan Ali, Data curation, Investigation, Project administration, Writing—review and editing; Alan E Hubbard, Resources, Supervision, Methodology, Project administration, Writing—review and editing; Christine P Stewart, Supervision, Funding acquisition, Methodology, Project administration, Writing—review and editing; Abul K Shoab, Md Ziaur Rahman, Md Saheen Hossen, Palash Mutsuddi, Syeda L Famida, Salma Akther, Data curation, Investigation, Project administration; Mahbubur

Rahman, Resources, Supervision, Funding acquisition, Investigation, Project administration, Writing—review and editing; Leanne Unicomb, Resources, Supervision, Funding acquisition, Project administration, Writing—review and editing; Firdaus S Dhabhar, Lia C H Fernald, Funding acquisition, Writing—review and editing; John M Colford Jnr, Stephen P Luby, Resources, Supervision, Funding acquisition, Methodology, Project administration, Writing—review and editing

### Author ORCIDs
Audrie Lin http://orcid.org/0000-0002-3877-3469
Benjamin F Arnold http://orcid.org/0000-0001-6105-7295

### Ethics
Clinical trial registration: The trial was registered at ClinicalTrials.gov (NCT01590095).
Human subjects: Primary caregivers of all children provided written informed consent. The study protocols were approved by human subjects committees at icddr,b (PR-11063 and PR-14108), the University of California, Berkeley (2011-09-3652 and 2014-07-6561) and Stanford University (25863 and 35583).

### Decision letter and Author response
Decision letter https://doi.org/10.7554/eLife.29365.sa1
Author response https://doi.org/10.7554/eLife.29365.sa2

## Additional files

### Supplementary files
- Reporting standard 1
- Transparent reporting form

### Data availability
The following dataset was generated:

| Author(s) | Year | Dataset title | Dataset URL | Database and Identifier |
|---|---|---|---|---|
| Lin A, Mertens A | 2017 | WASH Benefits Bangladesh Analysis of Telomere Outcomes | http://dx.doi.org/10.17605/OSF.IO/EVC98 | Publicly available at the Open Science Framework (project no. evc98), 10.17605/OSF.IO/EVC98 |

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
