## [Decision Letter]

Thank you for submitting your article "Effects of an early life intervention on telomere length among children in Bangladesh" for consideration by *eLife*. Your article has been reviewed by three peer reviewers, and the evaluation has been overseen by Eduardo Franco as Reviewing Editor and Prabhat Jha as the Senior Editor.

The reviewers have discussed the reviews with one another and the Reviewing Editor has drafted this decision to help you prepare a revised submission. Comments from the reviewers were edited for brevity but the first-person form was kept whenever useful to emphasize a point. Please note that the main points in the compiled critique below are grouped by heading but they may imply corrections or additions at different sections in the revised manuscript.

Summary:

This manuscript presents the results of a randomized controlled trial of a nutrition and sanitation intervention in Bangladesh, examining its effects on telomere length and telomere attrition. The results are intriguing and suggest telomere length may be affected by healthy interventions in the first years of life and lead to improved growth. The central element of the investigation was a relatively large (n=713) longitudinal cohort substudy based on multiple hygiene measures in impoverished households. The authors compared white blood cell (WBC) telomere length (T/S ratio) at 2 time points in young children from control and intervention households. Linear growth of the intervention group improved. Relative to controls, WBC T/S ratio was shorter after 1 year of intervention and similar between groups after 2 years. Between the 2 time points T/S ratio increased in the intervention group and decreased in the controls, a significant difference. In contrast with their hypotheses and other literature, results support an association between rapid growth (reflecting better health) in very early life and telomere shortening.

Essential revisions:

The overarching concerns that should orient the revision are:

1) There should be significant re-framing of the Introduction (i.e., this is a nutrition/sanitation intervention but the authors primarily discuss "adversity").

2) There should be additional analyses to investigate differential loss to follow-up, which is a major source of bias.

3) The results should be reframed as primarily null, emphasizing the more robust models rather than the unadjusted results.

4) More detail must be added to the Materials and methods section regarding blood processing, calculation of telomere length (as absolute base pairs – which are short for newborns/infants, etc.).

5) Discussion on potential mechanisms for the findings, including sources of bias.

Title:

The title as presented is not very descriptive. It should specify that the study is a nutrition and sanitation intervention, as opposed to generically stating "early life" intervention.

Methodological concerns relative to study design:

1) There needs to be more description of the intervention up front, especially since the results may be heavily influenced by the mechanisms through which the intervention is expected to operate. If not in the Results, then at the end of the Introduction. Otherwise it's hard to understand why such an intervention would influence TL and how.

2) It is unclear from the wording whether the control arm is group (1) or an entirely different group that received no intervention. (Results, first paragraph). (In the Materials and methods, a 7th control arm is a listed, which might be a better way to describe it here.)

3) Why weren't the other subjects in the intervention arms included for telomere analysis?

4) Why the focus on adversity? Are there other studies linking TL with nutrition, sanitation, infectious disease, or other exposures more relevant to this intervention?

5) Does this analysis only include children who were interviewed in both Time 1 and Time 2? If so, what percent of the total number of children do these represent? (I.e., how much overlap was there between the 66.5% in wave 1 and the 69.8% in wave 2?)

6) The Introduction suggests that TL may play a causal role in the etiology of disease (Introduction, end of first paragraph), although current evidence does not suggest that this is necessarily the case since most studies are associational. It's important to note that there are questions as to whether TL is simply associated with disease outcomes or actually a cause, particularly in recent Mendelian randomization studies, e.g., Cui et al., Atherosclerosis 2014; Hamad and Rehkopf, Experimental Gerontology 2016; Zhan et al., JAMA Neurology 2015. Perhaps soften this language and add to the discussion of study implications, i.e., we don't really know what the long-term causal implications are for shorter or longer telomeres.

7) Figure 1: Why were new children added to the treatment arm?

8) It's not clear how long the intervention operated, e.g., how long health promoters visited the compounds, whether the structural changes continued to be available throughout. I know that there is a citation to another paper that describes it in more detail, but we need a bit more detail here to understand why this intervention might influence TL.

Methodological concerns relative to measurement of telomere length:

1) More information and data should be included on how the telomere length (in bps) was calculated. 6763 bp is short for infants.

2) Has it been validated to wait so long from sample collection to measurement of TL (subsection “Procedures”, last paragraph)? If not, this is a limitation.

3) Were there differences in the month in which the treatment and control groups were measured? If so, there are known seasonal differences in TL at leukocytes at least (Rehkopf et al., AJHB 2014). Please discuss.

4) The blood sample for telomere length (by PCR) was stated to be lithium heparin, yet heparin inhibits PCR reactions. Sometimes this can be overcome by use of heparinise. Can the authors please clarify the type of specimen used, and if it was Li-heparin how this may have impacted the key results.

5) How measuring TL in leucocytes might influence the results, if the proposed mechanism of the intervention is largely infectious or nutritional? This should be listed as a limitation also, since TL measured in different tissues might yield different results based on a given intervention's mechanisms.

Concerns related to analysis and interpretation:

1) A critical issue is that unadjusted results are presented as the primary, and it is stated that the adjusted and IPCW results are "similar" (Results, fourth paragraph). This is not actually the case. While the point estimates are similar, they are no longer statistically significant, for the most part, in the adjusted and IPCW models. It would be more appropriate to conclude that this study showed less strong effects of the intervention on TL, than the authors imply. It seems inappropriate to consider the unadjusted analysis as primary, since so much differential loss to follow-up occurred.

2) Table 1 does not present the results for whether baseline covariates were statistically significantly different from one another for the two groups. Were there differences in sociodemographic characteristics between the two groups at Y1 and Y2? I.e., differential loss to follow-up that might explain the differences in TL and attrition? This is presented in a table but should be discussed here.

3) Many more children withdrew from the control group. This again highlights the need to present study characteristics to examine possible differential loss to follow-up. And this should be included as a major limitation, in that unmeasured characteristics may be driving these differences and the subsequent results. It also suggests that perhaps the adjusted analyses should be the primary ones.

4) Again, it would be more appropriate to frame this as a study that found null effects except for Year 1 differences, and to discuss why there may have been no effect observed. Moreover, there is not enough discussion of the hypothesized pathways linking adversity and TL, nor on how this intervention addresses adversity as opposed to nutrition and sanitation, which are very different exposures.

5) The authors should postulate mechanisms by which telomere shortening improves growth.

6) If telomere length shortening is associated with better immune functioning and the infants had improved hygienic conditions, then it is counterintuitive when considered in context with germ theory of disease. This should be amended and discussed.

7) The authors discuss adversity, which conjures to mind poverty, psychosocial stress, etc. It is inappropriate to talk about adversity because it was not measured via socioeconomic or psychosocial risk factors, so much as nutrition and sanitation. Please clarify what "adversity" means in these prior studies, and how/whether that relates to this particular intervention.

8) The limitations section needs to be expanded, e.g., lack of process evaluation measures that might clarify mechanistic pathways linking intervention to disease; limited understanding of how different types of exposures might influence TL; differential loss-to-follow-up on unmeasured characteristics that can't be accounted for by IPCW; capturing TL only in a single tissue. What are the implications of the lack of geographic matching for the substudy? What are the implications for generalizability and bias of the exclusion criteria, which eliminated potentially more disadvantaged individuals and therefore may have selected on better outcomes?

9) Only including "significantly related covariates" is not an appropriate way to select covariates. You should include all of those that are theoretically potential confounders. I.e., covariates should be selected conceptually and not empirically. It doesn't seem necessary to include (1) just age child age and sex; just do one fully adjusted model with all relevant covariates (subsection “Statistical parameters”, second paragraph).

10) The authors state that they conducted inverse probability weighting, but do not give these details (subsection “Missing outcomes”).

11) For both Tables 1 and 2, it's confusing to have mean (SD) and N (%) in the same column, and the tables overall are clutters with all the "%" signs. Perhaps have mean (SD) for continuous outcomes and just the% for binary outcomes?

12) Table 1: Is this table for the children who had telomere outcomes measured? Please specify in the footnote.

13) Table 1: At the top left, the table says "No. of compounds." Are the "N =" at the top of the second and third columns therefore number of compounds or number of children? If number of compounds, please present number of children also/instead. Similarly for the top of Table 2.

14) Tables 3 and 4 are very redundant. Can't you just add the IPCW column from Table 4 to Table 2? Perhaps put the CI below the point estimate so it can all fit?

15) The footnotes of Tables 3 and 4 say that covariates were pre-specified, but the text says that adjustment included significantly related covariates at p < 0.2. Please clarify.

16) Table 5 would look better if girls and boys were listed side by side instead of all in the same column. And actually, presenting it stratified in this way doesn't give us the information we want, which is the point estimate and CI for the interaction term of gender x treatment group. Are the headings for the mean and N columns reversed?

17) It is not intuitive why the argument for TL measurement is strengthened as a "potential target for future intervention development and evaluation." To the contrary, this suggests that the simple belief that longer TL = healthier needs to be further nuanced, and that additional studies like this one need to be done. We need a better understanding of the human physiology linking different types of socioeconomic and biomedical exposures with different types of telomere-related outcomes, since the existing literature suffers from too much confounding to understand what is actually going on. I'd say the main takeaway from this study is that we have a very limited understanding of how TL in humans actually responds to different types of interventions, and limited understanding of how TL actually influences different aspects of human health.

18) The presentation of the results in the Abstract is confusing, and doesn't parallel the presentation in the main manuscript. It would be helpful if the authors stated (1) the difference in TL between treatment and control at year 1; (2) the difference in TL between treatment and control at year 2; and (3) the difference in telomere attrition between treatment and control at year 1 and at year 2. These are the primary outcomes, as I understand them.

19) Unclear wording: does linear growth refer to telomeres or the child's height/length? (Abstract) If the latter, then this was not examined in this manuscript and should not go in the Abstract.

---

## [Author Response]

Essential revisions:The overarching concerns that should orient the revision are:1) There should be significant re-framing of the Introduction (i.e., this is a nutrition/sanitation intervention but the authors primarily discuss "adversity").

Thank you for this helpful recommendation. We have significantly revised and re-framed the Introduction section. We also added two new paragraphs. One paragraph focuses on the mechanisms by which the water, sanitation, and handwashing intervention could potentially affect TL: “Acute or chronic infections may contribute to childhood TL attrition, and inflammation and oxidative stress may be potential mediators (Houben, Moonen, van Schooten, and Hageman, 2008). […] To our knowledge, no studies have examined the impact of WSH interventions on TL.”

The other paragraph focuses on the mechanisms by which the nutritional intervention could potentially affect TL: “Early life nutrition may affect childhood TL. […] To our knowledge, no randomized controlled trials have assessed the effect of nutritional interventions on child TL.”

2) There should be additional analyses to investigate differential loss to follow-up, which is a major source of bias.

We agree with the reviewers that differential loss to follow-up could be a source of bias in this trial. This is an issue we had anticipated from the trial’s conception and included in our pre-specified and pre-registered analysis plans for the trial (all outcomes, not just telomere outcomes). In our original submission to *eLife* we included results from an inverse probability of censoring weighted (IPCW) analysis, which re-weights the analysis population to reflect the enrollment population using measured characteristics. The IPCW approach follows best practices for removing potential bias owing to missing outcome measurements in trials, according to a definitive methodologic review conducted by the National Research Council (Little et al., 2012). In this substudy, the IPCW analysis led to very similar inference as the adjusted and unadjusted analyses (original submission Table 4, revised submission Table 3), which suggests that outcomes were missing at random with respect to measurable characteristics and the missing outcomes are not a likely source of bias in the study. We made the following revisions to help emphasize the IPCW methods and results, and to help readers more clearly evaluate and interpret this sensitivity analysis in light of our primary findings:

-To address whether there were differences in enrollment characteristics between those with TL outcomes at Y1 versus Y2, we revised Table 1 to include treatment group means by year of measurement.

-As recommended by the reviewers, we combined the original Tables 3 and 4 into one table by adding the IPCW column from Table 4 to Table 3. This change enables readers to more clearly interpret the IPCW analysis beside the unadjusted and adjusted models.

-We have added more details on IPCW analysis to the Methods section: “The inverse probability of censoring weighted (IPCW) approach follows best practices for removing potential bias owing to missing outcome measurements in trials, according to a definitive methodologic review conducted by the National Research Council (Little et al., 2012). […] TL differences between the control and intervention group were then estimated using adjusted linear regression weighted by the inverse probability of TL measurement.”

-In the Discussion section, we added this language to the limitations paragraph: “Differential loss to follow-up in the control and intervention arms could potentially introduce bias if differential loss to follow-up was caused by unmeasured characteristics. However, our IPCW analysis showed that bias from loss to follow-up was unlikely based on a large set of measurable characteristics.”

3) The results should be reframed as primarily null, emphasizing the more robust models rather than the unadjusted results.

We agree with the reviewers and have reframed the Results and Discussion accordingly.

Typically, in a clinical trial, adjusted analyses (including IPCW) are pursued for two main reasons: 1) to remove bias from chance imbalances in group characteristics despite randomization or differential loss to follow-up, and 2) to increase the precision of the effect estimate (Pocock, Assmann, Enos, and Kasten, 2002). Balancing these objectives is a classic bias-variance tradeoff in statistics, and our approach, pre-specified in our analysis plan (https://osf.io/cjjwa/), has been to focus first on the issue of bias: namely, do we see evidence for any change in effect after adjustment? Then, if we see no evidence for bias, focus on the unadjusted analysis for our inference. This emphasis follows guidelines from leading statisticians in clinical trials, for example:

“Experiments offer more reliable evidence on causation than observational studies, which is not to gainsay the contribution to knowledge from observation. Experiments should be analyzed as experiments, not as observational studies. A simple comparison of rates might be just the right tool, with little value added by ‘sophisticated’ models.”

Freedman DA. Statistical models for causation – What inferential leverage do they provide? Eval Rev. 2006;30: 691–713.

“Experience shows that for most clinical trials, analyses which adjust for baseline covariates are in close agreement with the simpler unadjusted treatment comparisons. This is because (a) the randomization usually results in well balanced treatment groups, and (b) most covariates are not strongly related to the outcome.”

Pocock SJ, Assmann SE, Enos LE, Kasten LE. Subgroup analysis, covariate adjustment and baseline comparisons in clinical trial reporting: current practice and problems. Stat Med. 2002;21: 2917–2930.

As the reviewer notes below in comment #1 “Concerns related to analysis and interpretation”, the point estimates across analyses are very similar, which we find reassuring because it is consistent with good balance from randomization and outcome measurements missing at random (with respect to measurable characteristics). As the reviewer also notes, the confidence intervals widen slightly (and thus P-values increase slightly) as we move from the unadjusted analysis to the adjusted analyses. An analysis will only gain precision if all covariates are very strongly associated with the outcome (Pocock et al. 2002). Although our pre-specified algorithm pre-screened covariates that were associated with the outcome (a necessary condition for confounding), the adjusted analyses still lost a small amount of precision – suggesting that our conservative inclusion to reduce all possible bias led to a small loss in precision in this study. Given the similarity of the point estimates across analyses (Table 3), we feel that the unadjusted results, which rely only on randomization, are the most parsimonious summary of the trial.

We summarized our rationale in the Materials and methods section of our original submission: “Randomization led to balance in observed covariates across arms, so, in accordance with our pre-specified analysis plan, we relied on the unadjusted analysis as our primary analysis for the TL outcomes.”Tables 1 and 2 provide evidence that the groups with Y1 and Y2 TL outcomes were balanced across enrollment characteristics and that the group lost to follow up did not differ in enrollment characteristics from those with outcomes.

In the Results section, we addressed the reviewer’s concerns regarding the specific statement about unadjusted, adjusted, and IPCW analyses being “similar”. We clarified this statement for the change in TL from Y1 to Y2 result by emphasizing which model was significant: “Unadjusted, adjusted, and inverse probability of censoring weighting analyses generated similar estimates but only the unadjusted model was significant (Table 3).”Although we stated that the difference between the control and intervention arms in the change in relative TL from Y1 to Y2 was borderline significant (in the Results section), we agree that this borderline significant result is overemphasized in the Discussion section. Therefore, in the Discussion section, we have reframed this study as a study that found significant results at Y1, borderline significant/null effects in the difference in the relative change in TL between Y1 and Y2 among the two groups, and null effects at Y2.

To emphasize this point, we have added the borderline significant effects in the difference in the relative change in TL between Y1 and Y2 among the two groups and the null effects at Y2 to the first paragraph of the Discussion section: “The difference between the intervention and control arms in the change in relative TL from Y1 to Y2 was borderline significant, and at Y2, we found no difference in relative TL between the arms.”

In accordance with the guidelines for the interpretation of borderline significant results in clinical trials (Hackshaw and Kirkwood, 2011), we have softened the language of our interpretation and put forth alternative explanations.

The revised language in the Discussion emphasizes the borderline aspect of the change in TL result and enables the reader to interpret whether or not the difference is biologically meaningful and/or if the mechanisms proposed are plausible: “After the initial period of rapid linear growth during Y1, our borderline significant finding that telomere length increased in the intervention arm and decreased in the control arm from Y1 to Y2 seems to represent a preview of the TL trajectories that are set: we hypothesize that the intervention children will experience slower lifetime TL attrition and more physiological resistance to stress-related diseases (Juster, McEwen, and Lupien, 2010). […] The smaller than expected treatment effect and the borderline p-value indicate that the trial was slightly underpowered to detect differences in the change in relative TL from Y1 to Y2 among the intervention and control groups.”

Additionally, we added a new paragraph with an alternative interpretation if the reader considers the borderline result to be null: “Alternatively, our finding of a modest, borderline impact from Y1 to Y2 may suggest that the effect of the interventions is diminished in children over 1 year of age. […] This interpretation would also be consistent with no differences in relative TL observed at Y2 between the two groups, underscoring the importance of targeting interventions early in life during the sensitive period when they are likely to have the largest impact on childhood TL.”

4) More detail must be added to the Materials and methods section regarding blood processing, calculation of telomere length (as absolute base pairs – which are short for newborns/infants, etc.).

We added a paragraph to the Materials and methods section with these details (below).

The new paragraph in the Materials and methods section reads: “To determine the conversion factor for the calculation of approximate base pair telomere length from T/S ratio, the above method was used to determine the T/S ratios, relative to the same reference DNA, for a set of genomic DNA samples from the ATCC authenticated human fibroblast primary cell line IMR-90 (ATCC: CCL^-^186, cell line authentication method: STR profiling; ATCC determined the cell line was free of mycoplasma contamination) at different population doublings, as well as with the telomerase protein subunit gene (hTERT) transfected into a lentiviral construct. […] Comparison of T/S ratios versus base pairs derived from the Southern blot analysis generated the following equation for conversion from T/S ratio to base pairs: base pairs = 3274 + 2413 * (T/S) (Farzaneh-Far et al., 2010).”

We agree that the averages of 6729 bp at Year 1 and 6763 bp at Year 2 are short for newborns compared to some previous findings (Factor-Litvak et al., 2016; Frenck, Blackburn, and Shannon, 1998; Wojcicki, Heyman, et al., 2016), but these averages are similar to a recent report of newborns whose mothers experienced a high level of stress during pregnancy (Marchetto et al., 2016). It is possible that the short telomere length in our cohort is due to the overall low socioeconomic status, which is shown to be associated with shorter telomere length, although we cannot rule out other factors that may contribute to this, such as race, malnutrition, or infection status.

We have added new language to the Results section: “Although these averages are short for young children compared to some previous findings (Factor-Litvak et al., 2016; Frenck, Blackburn, and Shannon, 1998; Wojcicki, Heyman, et al., 2016), they are similar to a recent report of newborns whose mothers experienced a high level of stress during pregnancy (Marchetto et al., 2016); the stress associated with low socioeconomic status may have contributed to the short average TL in this study.”

5) Discussion on potential mechanisms for the findings, including sources of bias.

We have expanded the discussion on potential mechanisms for the findings by adding a new paragraph to the Discussion section. Please see our response to comment #5 under “Concerns related to analysis and interpretation” for additional details (below).

The new paragraph reads: “The accelerated TL attrition during the first year of life in the intervention arm may also reflect better linear growth. […] Forthcoming WASH Benefits studies focused on child inflammation and enteric pathogen burden will provide evidence on mechanistic pathways leading to different TL, growth, and development outcomes.”

We also vastly expanded the limitations paragraph on sources of bias in the Discussion section. Please see our response to comment #8 under “Concerns related to analysis and interpretation” for additional details (below).

The revised limitations paragraph reads: “This study had several limitations. […] Finally, we did not measure TL at birth (Wojcicki, Olveda, et al., 2016), but we would expect it to be similar for both groups because household characteristics were balanced by randomization.”

Title:The title as presented is not very descriptive. It should specify that the study is a nutrition and sanitation intervention, as opposed to generically stating "early life" intervention.

The *eLife* guidelines require titles to be a maximum of 120 characters; therefore, we generated our original title to meet these guidelines.

As suggested, we have changed the title to “Effects of water, sanitation, handwashing, and nutritional interventions on telomere length among children in a cluster-randomized controlled trial in rural Bangladesh.” These changes result in a 147-character title, slightly exceeding the maximum character limit. We included “cluster-randomized controlled trial” in the title according to CONSORT guidelines for randomized controlled trials.

Methodological concerns relative to study design:1) There needs to be more description of the intervention up front, especially since the results may be heavily influenced by the mechanisms through which the intervention is expected to operate. If not in the Results, then at the end of the Introduction. Otherwise it's hard to understand why such an intervention would influence TL and how.

We added details of the intervention hardware to the description of the intervention in the Results section. Further details are included in the original Materials and methods section.

The revised sentence reads: “Between 31 May 2012 and 7 July 2013, in the overall trial, 5551 compounds (collections of related households) with pregnant women in their first or second trimester were randomly allocated to one of the six intervention groups or a double sized control group as follows: 1) chlorinated drinking water and safe storage vessel, 2) upgraded sanitation (child potties, sani-scoop hoes to remove feces, and a double pit latrine with a hygienic water seal), 3) handwashing promotion (handwashing stations with detergent soap), 4) combined water + sanitation + handwashing 5) nutrition (lipid-based nutrient supplements and age-appropriate recommendations on maternal nutrition and infant feeding practices), 6) combined nutrition + water + sanitation + handwashing (N+WSH), 7) control group, which did not receive any interventions (Figure 1). Community health promoters visited study compounds in the intervention arms to promote behaviors.”

2) It is unclear from the wording whether the control arm is group (1) or an entirely different group that received no intervention. (Results, first paragraph). (In the Materials and methods, a 7th control arm is a listed, which might be a better way to describe it here.)

The control arm received no intervention – we have clarified this in the revision following the reviewer’s suggestion. Please see our response to comment #1 under “Methodological concerns relative to study design” for additional details (above).

3) Why weren't the other subjects in the intervention arms included for telomere analysis?

Venipuncture samples were only collected from children in the control arm, the nutrition arm, the combined water + sanitation + handwashing (WSH) arm, and the combined N+WSH arm. Originally, we proposed telomere analysis in all four of these arms. However, we were not able to obtain sufficient funding to include the other intervention arms (nutrition arm and combined WSH arm) in the telomere analysis. After the publication of these results, we hope to obtain additional funding for a future study on the banked biological specimens that includes the other intervention arms (the individual nutrition arm and the combined WSH arm). This future study would determine if the observed differences in telomere length at Year 1 between the control and combined N+WSH arms were due to nutrition alone, combined WSH only, or the combination of N+WSH.

4) Why the focus on adversity? Are there other studies linking TL with nutrition, sanitation, infectious disease, or other exposures more relevant to this intervention?

We have added two new paragraphs to the Introduction section (please refer to our response to comment #1 in ‘Essential revisions’ above for details) focused on background literature related to the interventions. The new paragraphs include adult studies focused on infections, infectious diseases, and multivitamin supplementation. We originally focused on adversity because the few studies conducted in children measured TL in the context of adversity. To date, the studies linking TL with nutrition and infectious disease were mostly focused on adults in high-income countries. We do not know of any studies linking TL with handwashing, drinking water, or sanitation interventions.

5) Does this analysis only include children who were interviewed in both Time 1 and Time 2? If so, what percent of the total number of children do these represent? (I.e., how much overlap was there between the 66.5% in wave 1 and the 69.8% in wave 2?)

It is unclear to which analysis this comment pertains to. Overall, the analysis at Year 1 includes children with TL measurements at Year 1 (Time 1). The 66.5% is the percentage of children with TL outcomes at Year 1 of the total number of children visited at Year 1 (662/996=66.5%). The analysis at Year 2 includes children with TL measurements at Y2 (Time 2). The 69.8% is the percentage of children with TL outcomes at Year 2 of the total number of children visited at Year 2 (713/1021=69.8%). The analysis of the change in TL from Year 1 to Year 2 only included children with TL outcomes at both time points (N=557). These 557 children represent 54.6% of the total children visited between Year 1 and Year 2 (557/1021). The number of children included in each analysis is displayed in Table 3.

We have added this percentage to the following sentence in the Results section: “Of index children with a TL measurement at both Y1 and Y2 (N=557; 54.6% of the total children visited), relative TL increased by +0.03 T/S (+76 bp) in the intervention group and decreased by -0.01 T/S (-23 bp) in the control group (Table 3).”

6) The Introduction suggests that TL may play a causal role in the etiology of disease (Introduction, end of first paragraph), although current evidence does not suggest that this is necessarily the case since most studies are associational. It's important to note that there are questions as to whether TL is simply associated with disease outcomes or actually a cause, particularly in recent Mendelian randomization studies, e.g., Cui et al., Atherosclerosis 2014; Hamad and Rehkopf, Experimental Gerontology 2016; Zhan et al., JAMA Neurology 2015. Perhaps soften this language and add to the discussion of study implications, i.e., we don't really know what the long-term causal implications are for shorter or longer telomeres.

Thank you for the recommendations and for the references. We completely agree with the reviewer’s comment about uncertainty as to the etiologic and causal relationship between telomeres and disease, and have followed the suggestion to soften the language of this sentence – also adding two of the suggested references (Hamad, Walter, and Rehkopf, Experimental Gerontology 2016 and Zhan et al., JAMA Neurology 2015).

The revised sentence reads:“It remains an open debate whether TL serves as a “molecular clock” that gauges cumulative stress exposures over a lifespan or plays a role in the etiology of various diseases (Blackburn, Epel, and Lin, 2015; Hamad, Walter, and Rehkopf, 2016; Zhan et al., 2015).”

We have revised this sentence in the Discussion section: “Future studies are needed to understand the environmental exposures contributing to TL dynamics during early life and the critical physiological pathways linking TL and lifelong health trajectories.”

7) Figure 1: Why were new children added to the treatment arm?

The original sample size for children enrolled in the intensive biological sampling was based on calculations for an environmental enteric dysfunction substudy. We sampled more children within the treatment arm to maintain sample size and power in the substudy since each environmental enteric dysfunction outcomes were analyzed at each separate time point, and we were sampling from the same study base. This telomere measurement was nested within the environmental enteric dysfunction substudy, and we analyzed any available biological samples in the control and N+WSH arms.

To clarify, we changed the language in Figure 1 – Follow-up to “New children measured”.

8) It's not clear how long the intervention operated, e.g., how long health promoters visited the compounds, whether the structural changes continued to be available throughout. I know that there is a citation to another paper that describes it in more detail, but we need a bit more detail here to understand why this intervention might influence TL.

The interventions operated for 2 years in total.

To address the reviewer’s comment, we have revised the first sentence of the first paragraph in the Results section: “The objective of the WASH Benefits trial was to compare the effects of individual and combined interventions on child health in the first two years of life – the critical window to prevent growth faltering (Arnold et al., 2013).”

Please also refer to the first paragraph of the Results section:“The substudy team visited 996 index children after 1 year of intervention (Y1) and 1021 children after 2 years of intervention (Y2) in the control and N+WSH arms only.”

We have revised this sentence in the Materials and methods section to address the duration of health promoter visits: “Community health promoters visited study compounds in the intervention arms at least once per week during the first 6 months and at least once biweekly to promote behaviors.”

We have revised this sentence in the Results section to address the duration of the observed structural changes: “Measures of intervention adherence (including observed hardware availability) were high (over 80%) and sustained throughout Y1 and Y2 (S. P. Luby et al. in review).”

Methodological concerns relative to measurement of telomere length:1) More information and data should be included on how the telomere length (in bps) was calculated. 6763 bp is short for infants.

We have added the requested information and data to the Materials and methods section, and discussed the short infant TL. Please see our response to comment #4 under “Essential revisions” for details.

2) Has it been validated to wait so long from sample collection to measurement of TL (subsection “Procedures”, last paragraph)? If not, this is a limitation.

We have not assessed the impact of blood storage time on telomere length measurement, although the majority of the published literature on telomere length in human studies used archived blood (Eisenberg et al., 2017). In the Eisenberg et al. study, the duration of time between venous blood collection and TL measurement was 7 years (Eisenberg et al., 2017). In contrast, in our study, the duration of time from sample collection to TL measurement ranged from 8 to 32 months. We have added it as a limitation to the Discussion section.

The new language reads: “The lithium heparin additive in the whole blood may have inhibited the qPCR reaction and the long duration of time from sample collection to TL measurement may have affected TL, but any systematic measurement errors would likely affect both study groups equally. These factors may provide potential explanations for the shorter average TL observed in this study compared to other studies (Factor-Litvak et al., 2016; Frenck et al., 1998; Wojcicki, Heyman, et al., 2016).”

3) Were there differences in the month in which the treatment and control groups were measured? If so, there are known seasonal differences in TL at leukocytes at least (Rehkopf et al., AJHB 2014). Please discuss.

When we plot TL versus month of data collection for the treatment and control groups (see Author response image 1), we do not see evidence of seasonal changes. In the fully adjusted model, one of the pre-specified covariates was month of data collection to account for seasonal changes.

We have added new language to the Results section: “Although a previous study reported seasonal variation in TL (Rehkopf et al., 2014), we did not observe seasonal changes within this study.”

4) The blood sample for telomere length (by PCR) was stated to be lithium heparin, yet heparin inhibits PCR reactions. Sometimes this can be overcome by use of heparinise. Can the authors please clarify the type of specimen used, and if it was Li-heparin how this may have impacted the key results.

The original text in the Materials and methods section states the type of specimen used: “For each child, trained icddr,b staff collected a 5 ml venipuncture sample from the antecubital area of the arm into a Sarstedt S-monovette lithium heparin collection tube […] Genomic DNA was extracted from heparin-anti-coagulated whole blood stored at -80°C using QIAamp DNA Mini Kit (QIAGEN, Hilden, Germany).”

We did not use heparinase in the PCR reaction. The results that may have been affected by the lithium heparin are the relative telomere lengths for each individual. Since lithium heparin tubes were used for the collection of all blood samples, we would expect any impact of the heparin on the qPCR reaction to affect both the control and intervention groups similarly. Therefore, we would not expect the key differences between the control and intervention groups to be impacted by the heparin.

We added this heparin issue as a potential limitation to the Discussion section. The new sentence reads: “The lithium heparin additive in the whole blood may have inhibited the qPCR reaction and the long duration of time from sample collection to TL measurement may have affected TL, but any systematic measurement errors would likely affect both study groups equally. These factors may provide potential explanations for the shorter average TL observed in this study compared to other studies (Factor-Litvak et al., 2016; Frenck et al., 1998; Wojcicki, Heyman, et al., 2016).”

5) How measuring TL in leucocytes might influence the results, if the proposed mechanism of the intervention is largely infectious or nutritional? This should be listed as a limitation also, since TL measured in different tissues might yield different results based on a given intervention's mechanisms.

Thank you for raising this issue. If the mechanism of the intervention is through the promotion of better immune development, we would expect to observe differences in leukocyte TL (e.g., due to T-cell proliferation), but not in less proliferative tissues such as muscle. We added this issue to the limitations paragraph of the Discussion section.

The new language reads: “We only measured TL in whole blood, which might yield different results compared to TL measured in less proliferative tissue types (e.g., muscle or fat) (Daniali et al., 2013).”

Concerns related to analysis and interpretation:1) A critical issue is that unadjusted results are presented as the primary, and it is stated that the adjusted and IPCW results are "similar" (Results, fourth paragraph). This is not actually the case. While the point estimates are similar, they are no longer statistically significant, for the most part, in the adjusted and IPCW models. It would be more appropriate to conclude that this study showed less strong effects of the intervention on TL, than the authors imply. It seems inappropriate to consider the unadjusted analysis as primary, since so much differential loss to follow-up occurred.

[repeated from Essential revisions comment #3 response, above]Typically, in a clinical trial, adjusted analyses (including IPCW) are pursued for two main reasons: 1) to remove bias from chance imbalances in group characteristics despite randomization or differential loss to follow-up, and 2) to increase the precision of the effect estimate (Pocock et al., 2002). Balancing these objectives is a classic bias-variance tradeoff in statistics, and our approach, pre-specified in our analysis plan (https://osf.io/cjjwa/), has been to focus first on the issue of bias: namely, do we see evidence for any change in effect after adjustment? Then, if we see no evidence for bias, focus on the unadjusted analysis for our inference. This emphasis follows guidelines from leading statisticians in clinical trials, for example:

“Experiments offer more reliable evidence on causation than observational studies, which is not to gainsay the contribution to knowledge from observation. Experiments should be analyzed as experiments, not as observational studies. A simple comparison of rates might be just the right tool, with little value added by ‘sophisticated’ models.”

Freedman DA. Statistical models for causation – What inferential leverage do they provide? Eval Rev. 2006;30: 691–713.

“Experience shows that for most clinical trials, analyses which adjust for baseline covariates are in close agreement with the simpler unadjusted treatment comparisons. This is because (a) the randomization usually results in well balanced treatment groups, and (b) most covariates are not strongly related to the outcome.”

Pocock SJ, Assmann SE, Enos LE, Kasten LE. Subgroup analysis, covariate adjustment and baseline comparisons in clinical trial reporting: current practice and problems. Stat Med. 2002;21: 2917–2930.

As the reviewer notes, the point estimates across analyses are very similar, which we find reassuring because it is consistent with good balance from randomization and outcome measurements missing at random (with respect to measurable characteristics). As the reviewer also notes, the confidence intervals widen slightly (and thus P-values increase slightly) as we move from the unadjusted analysis to the adjusted analyses. An analysis will only gain precision if all covariates are very strongly associated with the outcome (Pocock et al. 2002). Although our pre-specified algorithm pre-screened covariates that were associated with the outcome (a necessary condition for confounding), the adjusted analyses still lost a small amount of precision – suggesting that our conservative inclusion to reduce all possible bias led to a small loss in precision in this study. Given the similarity of the point estimates across analyses (Table 3), we feel that the unadjusted results, which rely only on randomization, are the most parsimonious summary of the trial.

We summarized our rationale in the Materials and methods section of our original submission: “Randomization led to balance in observed covariates across arms, so, in accordance with our pre-specified analysis plan, we relied on the unadjusted analysis as our primary analysis for the TL outcomes.”Tables 1 and 2 provide evidence that the groups with Y1 and Y2 TL outcomes were balanced across enrollment characteristics and that the group lost to follow up did not differ in enrollment characteristics from those with outcomes.

In the Results section, we addressed the reviewer’s concerns regarding the specific statement about unadjusted, adjusted, and IPCW analyses being “similar”. We clarified this statement for the change in TL from Y1 to Y2 result by emphasizing which model was significant: “Unadjusted, adjusted, and inverse probability of censoring weighting analyses generated similar estimates but only the unadjusted model was significant (Table 3).”Although we stated that the difference between the control and intervention arms in the change in relative TL from Y1 to Y2 was borderline significant (in the Results section), we agree that this borderline significant result is overemphasized in the Discussion section. Therefore, we have softened the language and reframed the discussion of this result in the Discussion section in accordance with guidelines for the interpretation of borderline significant results within clinical trials (Hackshaw and Kirkwood, 2011). Please see our response to comment #4 under “Concerns related to analysis and interpretation” for the revised language in the Discussion section.

2) Table 1 does not present the results for whether baseline covariates were statistically significantly different from one another for the two groups. Were there differences in sociodemographic characteristics between the two groups at Y1 and Y2? I.e., differential loss to follow-up that might explain the differences in TL and attrition? This is presented in a table but should be discussed here.

In a clinical trial, any imbalance in baseline characteristics despite randomization is, by definition, due to chance. The issue was best summarized by Doug Altman:

“In a clinical trial in which treatment allocation was properly randomized, a difference of any sort between the two groups at the time of entry to the trial will necessarily be due to chance, since randomization precludes any external influences (biases) on which subjects receive which treatments […] Performing a significance test to compare baseline variables is to assess the probability of something having occurred by chance when we know that it did occur by chance”.

Altman DG. Comparability of Randomised Groups. Statistician. 1985;34: 125–136.

Instead, Altman recommends displaying enrollment characteristics of the groups in a table to enable the reader to assess balance subjectively. Subjective assessment requires “prior knowledge of the prognostic importance of the variables being compared”. (Altman, 1985)

To address whether there were differences in enrollment characteristics between those with TL outcomes at Y1 versus Y2, we revised Table 1 to include treatment group means by year of measurement.

[repeated from Essential revisions comment #2 response, above] We agree with the reviewers that differential loss to follow-up could potentially be a source of bias in this trial. This is an issue we had anticipated from the trial’s conception and included in our pre-specified and pre-registered analysis plans for the trial (all outcomes, not just telomere outcomes). In our original submission to *eLife* we included results from an inverse probability of censoring weighted (IPCW) analysis, which re-weights the analysis population to reflect the enrollment population using measured characteristics. The IPCW approach follows best practices for removing potential bias owing to missing outcome measurements in trials, according to a definitive methodologic review conducted by the National Research Council (Little et al., 2012). In this study, the IPCW analysis led to very similar inference as the adjusted and unadjusted analyses (original submission Table 4, revised submission Table 3), which suggests that outcomes were missing at random with respect to measurable characteristics and the missing outcomes are not a likely source of bias in the study.

In the original submission, we included Table 2 featuring the enrollment characteristics of households with TL measurement at Year 1 versus those lost to follow-up at Year 2. We have added differential loss to follow-up as a potential limitation in the Discussion section.

The new language in the Discussion section reads: “Differential loss to follow-up in the control and intervention arms could potentially introduce bias if differential loss to follow-up was caused by unmeasured characteristics. However, our IPCW analysis showed that bias from loss to follow-up was unlikely based on a large set of measurable characteristics.”

3) Many more children withdrew from the control group. This again highlights the need to present study characteristics to examine possible differential loss to follow-up. And this should be included as a major limitation, in that unmeasured characteristics may be driving these differences and the subsequent results. It also suggests that perhaps the adjusted analyses should be the primary ones.

Please see our response to comment #2 “Concerns related to analysis and interpretation” above.

Thank you for raising this issue. To address the differential loss to follow-up comment, we added this language to the Discussion section: “Differential loss to follow-up in the control and intervention arms could potentially introduce bias if differential loss to follow-up was caused by unmeasured characteristics. However, our IPCW analysis showed that bias from loss to follow-up was unlikely based on a large set of measurable characteristics.”

4) Again, it would be more appropriate to frame this as a study that found null effects except for Year 1 differences, and to discuss why there may have been no effect observed. Moreover, there is not enough discussion of the hypothesized pathways linking adversity and TL, nor on how this intervention addresses adversity as opposed to nutrition and sanitation, which are very different exposures.

[repeated from Essential revisions comment #3 response, above] In the Discussion section, we have reframed this study as a study that found significant results at Y1, borderline significant/null effects in the difference in the relative change in TL between Y1 and Y2 among the two groups, and null effects at Y2.

To emphasize this point, we have added the borderline significant effects in the difference in the relative change in TL between Y1 and Y2 among the two groups and the null effects at Y2 to the first paragraph of the Discussion section: “The difference between the intervention and control arms in the change in relative TL from Y1 to Y2 was borderline significant, and at Y2, we found no difference in relative TL between the arms.”

In accordance with the guidelines for the interpretation of borderline significant results in clinical trials (Hackshaw and Kirkwood, 2011), we have softened the language of our interpretation and put forth alternative explanations.

The revised language in the Discussion emphasizes the borderline aspect of the change in TL result and enables the reader to interpret whether or not the difference is biologically meaningful and/or if the mechanisms proposed are plausible: “After the initial period of rapid linear growth during Y1, our borderline significant finding that telomere length increased in the intervention arm and decreased in the control arm from Y1 to Y2 seems to represent a preview of the TL trajectories that are set: we hypothesize that the intervention children will experience slower lifetime TL attrition and more physiological resistance to stress-related diseases (Juster et al., 2010). […] The smaller than expected treatment effect and the borderline p-value indicate that the trial was slightly underpowered to detect differences in the change in relative TL from Y1 to Y2 among the intervention and control groups.”

Additionally, we added a new paragraph with an alternative interpretation if the reader considers the borderline result to be null: “Alternatively, our finding of a modest, borderline impact from Y1 to Y2 may suggest that the effect of the interventions is diminished in children over 1 year of age. […] This interpretation would also be consistent with no differences in relative TL observed at Y2 between the two groups, underscoring the importance of targeting interventions early in life during the sensitive period when they are likely to have the largest impact on childhood TL.”

We have amended the Discussion section to remove the focus on adversity (as suggested by the reviewer in comment #7 under “Concerns related to analysis and interpretation” below). Furthermore, we have expanded on our original discussion of mechanisms by including a new paragraph in the Discussion on the hypothesized pathways linking nutrition and WSH exposures and childhood TL and growth. Please see our response to comment #5 under “Concerns related to analysis and interpretation” (below) for more details.

5) The authors should postulate mechanisms by which telomere shortening improves growth.

In the original Discussion section, we hypothesized that telomere shortening may reflect improved growth (e.g., a biomarker of growth) and did not imply that telomere shortening would necessarily be considered a cause of improved growth (if anything, our hypothesis based on the results would imply causality in the other direction: improved growth → shorter telomeres). We have included language in the Discussion to further clarify our hypothesis (please see new text below). Our study was not designed to examine the potential causal pathway between telomere shortening and improved growth. Therefore, we prefer not to postulate mechanisms for a relationship that we have not yet examined or proposed.

Instead, to address this point and comment #5 under “Essential revisions” we have expanded the discussion on potential mechanisms for improved TL and growth observed in the intervention arm by adding this paragraph to the Discussion section: [repeated from Essential revisions comment #5 response, above] “The accelerated TL attrition during the first year of life in the intervention arm may also reflect better linear growth. […] Forthcoming WASH Benefits studies focused on child inflammation and enteric pathogen burden will provide evidence on mechanistic pathways leading to different TL, growth, and development outcomes.”

6) If telomere length shortening is associated with better immune functioning and the infants had improved hygienic conditions, then it is counterintuitive when considered in context with germ theory of disease. This should be amended and discussed.

We have not made this amendment because our interpretation is that the observed results and hypothesized relationship between hygienic conditions, immune function, and telomere length shortening are all internally consistent with the germ theory of disease. The germ theory of disease states that infectious diseases are caused by the presence of pathogenic microorganisms within the body. As discussed in the Discussion section, the children in the N+WSH intervention arm potentially experienced improved hygienic conditions, which likely limited exposure to microorganisms, as exhibited by the reductions in caregiver-reported diarrhea, soil-transmitted helminth infections, and *Giardia duodenalis* infections (S. P. Luby et al., A. Ercumen et al., and A. Lin et al. in review). The children in the N+WSH intervention arm likely experienced fewer infections and perhaps had better immune system development, which in turn was reflected by shorter telomeres.

7) The authors discuss adversity, which conjures to mind poverty, psychosocial stress, etc. It is inappropriate to talk about adversity because it was not measured via socioeconomic or psychosocial risk factors, so much as nutrition and sanitation. Please clarify what "adversity" means in these prior studies, and how/whether that relates to this particular intervention.

We significantly reframed the Introduction section by removing the focus on adversity and instead, adding two new paragraphs on the potential mechanisms by which nutrition and WSH could affect TL. For more details, please see our response to comment #1 under “Essential revisions” (above).

We have also amended the Discussion to clarify the term “adversity” in prior studies: “This result is contrary to many studies that reported increased TL attrition associated with prenatal psychosocial stress (Entringer et al., 2013; Marchetto et al., 2016), childhood institutional care (Drury et al., 2012), disease (Fitzpatrick et al., 2007; Salpea et al., 2010), and mortality (Cawthon, Smith, O'Brien, Sivatchenko, and Kerber, 2003); however, these prior studies did not examine participants in the same age range (1-2 years) as this study. In high- and low-resource settings, there is a dearth of evidence on the effect of environmental exposures on TL in this age range and its implications for adult health outcomes.”

8) The limitations section needs to be expanded, e.g., lack of process evaluation measures that might clarify mechanistic pathways linking intervention to disease; limited understanding of how different types of exposures might influence TL; differential loss-to-follow-up on unmeasured characteristics that can't be accounted for by IPCW; capturing TL only in a single tissue. What are the implications of the lack of geographic matching for the substudy? What are the implications for generalizability and bias of the exclusion criteria, which eliminated potentially more disadvantaged individuals and therefore may have selected on better outcomes?

Thank you for identifying these additional limitations. We have expanded the limitations section to discuss all of the reviewer’s suggestions except the following points: “lack of process evaluation measures that might clarify mechanistic pathways linking intervention to disease”, “limited understanding of how different types of exposures might influence TL”, and “bias of the exclusion criteria, which eliminated potentially more disadvantaged individuals and therefore may have selected on better outcomes”. The process evaluation measures are described in detail in the main trial publication (S. P. Luby et al. in review), and all analyses were intention-to-treat. The limited understanding of how different types of exposures might influence TL is not a limitation specific to this study, rather it is a limitation of the general TL literature. Therefore, we have included this point in the final two sentences of the Discussion section, but have not highlighted it in the limitations paragraph. The venipuncture exclusion criteria only excluded two children: one child in the control arm at Year 1 and one child in the intervention arm at Year 2. We added this information to the Materials and methods section. Since the number of children excluded is small, we have decided not to include the exclusion criteria as a limitation.

[repeated from Essential Revisions comment #5 response, above] The revised limitations paragraph reads: *“*This study had several limitations. […] Finally, we did not measure TL at birth (Wojcicki, Olveda, et al., 2016), but we would expect it to be similar for both groups because household characteristics were balanced by randomization.”

9) Only including "significantly related covariates" is not an appropriate way to select covariates. You should include all of those that are theoretically potential confounders. I.e., covariates should be selected conceptually and not empirically. It doesn't seem necessary to include (1) just age child age and sex; just do one fully adjusted model with all relevant covariates (subsection “Statistical parameters”, second paragraph).

Please see our response to comment #1 under “Concerns related to analysis and interpretation” for our broader perspective on adjusted analyses in the context of clinical trials. In regard to this specific point, by definition a covariate can confound a relationship between an exposure (here: randomized treatment) and the outcome if it is associated with the outcome (VanderWeele and Shpitser, 2011). However, it is widely known that covariates can hurt precision in an analysis if they are not strongly associated with the outcome. Therefore, this pre-screening step is a common approach for covariate selection in trials – and epidemiologic studies more broadly – to help avoid losing too much precision in the adjusted analysis through inclusion of covariates that are unassociated with the outcome (and thus could not be confounders). As long as the approach is algorithmic and pre-specified, it is not susceptible to “cherry picking”. Below is an excerpt of an influential article on the analysis of clinical trials that contributed to the development of our pre-specified approach to covariate selection in this trial:

“Some have argued that prespecification of all covariates for adjustment in a single model is the best solution, leading to just one predefined covariate-adjusted analysis [10]. […] While this means the chosen covariates themselves could not be prespecified, a precise predefined statistical strategy for variable selection should overcome somewhat any suspicions that post hoc selection of covariates might be based on subjective criteria.”

Pocock SJ, Assmann SE, Enos LE, Kasten LE. Subgroup analysis, covariate adjustment and baseline comparisons in clinical trial reporting: current practice and problems. Stat Med. 2002;21: 2917–2930.

10) The authors state that they conducted inverse probability weighting, but do not give these details (subsection “Missing outcomes”).

We have added more details on IPCW analysis to the Materials and methods section.

The newly added language reads: “The inverse probability of censoring weighted (IPCW) approach follows best practices for removing potential bias owing to missing outcome measurements in trials, according to a definitive methodologic review conducted by the National Research Council (Little et al., 2012). […] TL differences between the control and intervention group were then estimated using adjusted linear regression weighted by the inverse probability of TL measurement.”

11) For both Tables 1 and 2, it's confusing to have mean (SD) and N (%) in the same column, and the tables overall are clutters with all the "%" signs. Perhaps have mean (SD) for continuous outcomes and just the% for binary outcomes?

As suggested, we have displayed the mean (SD) for continuous outcomes and only the% for binary outcomes. We have retained the% signs for clarity.

12) Table 1: Is this table for the children who had telomere outcomes measured? Please specify in the footnote.

Yes, this table is for children who had telomere outcomes measured.

We have added this language to the footnote: “Enrollment characteristics of households with children who had telomere measurements.”

13) Table 1: At the top left, the table says "No. of compounds." Are the "N =" at the top of the second and third columns therefore number of compounds or number of children? If number of compounds, please present number of children also/instead. Similarly for the top of Table 2.

Yes, these numbers are the number of children.

In Table 1, we have added changed the heading to: “No. of children”

14) Tables 3 and 4 are very redundant. Can't you just add the IPCW column from Table 4 to Table 2? Perhaps put the CI below the point estimate so it can all fit?

We agree and have combined the original Tables 3 and 4 into one table by adding the IPCW column from Table 4 to Table 3.

15) The footnotes of Tables 3 and 4 say that covariates were pre-specified, but the text says that adjustment included significantly related covariates at p < 0.2. Please clarify.

We have clarified the footnote:“Adjusted for pre-specified covariates associated with the outcome (likelihood ratio test P-value <0.2).”

16) Table 5 would look better if girls and boys were listed side by side instead of all in the same column. And actually, presenting it stratified in this way doesn't give us the information we want, which is the point estimate and CI for the interaction term of gender x treatment group. Are the headings for the mean and N columns reversed?

We have made the changes requested and included the point estimate and CI for the interaction term. Please note that the original Table 5 has been renumbered as Table 4.

17) It is not intuitive why the argument for TL measurement is strengthened as a "potential target for future intervention development and evaluation." To the contrary, this suggests that the simple belief that longer TL = healthier needs to be further nuanced, and that additional studies like this one need to be done. We need a better understanding of the human physiology linking different types of socioeconomic and biomedical exposures with different types of telomere-related outcomes, since the existing literature suffers from too much confounding to understand what is actually going on. I'd say the main takeaway from this study is that we have a very limited understanding of how TL in humans actually responds to different types of interventions, and limited understanding of how TL actually influences different aspects of human health.

Thank you for synthesizing the salient points of our study. We strongly agree with the reviewer’s comments. Additionally, there are gaps in the literature on child TL dynamics in health and disease during the first two years of life. As suggested by the reviewer, we have revised and strengthened the concluding sentences of the Discussion based on these points.

The revised language reads: “Although TL was a sensitive outcome that responded to an intervention, this trial underscores our limited understanding of environmental exposures contributing to TL dynamics during early life and the critical physiological pathways linking TL and lifelong health trajectories. Since potential confounding could plague observational studies, evaluating the relationship between modifiable exposures and TL within the context of randomized controlled trials could provide valuable contributions to the field of telomere biology.”

18) The presentation of the results in the Abstract is confusing, and doesn't parallel the presentation in the main manuscript. It would be helpful if the authors stated (1) the difference in TL between treatment and control at year 1; (2) the difference in TL between treatment and control at year 2; and (3) the difference in telomere attrition between treatment and control at year 1 and at year 2. These are the primary outcomes, as I understand them.

We have restructured the Abstract to parallel the results in the main manuscript by adding point 2: difference in treatment and control at Year 2. The original Abstract already included points 1 and 3.

The revised paragraph reads: “Children in the intervention arm had significantly shorter relative TL compared with controls after 1 year of intervention (difference -163 bp, P=0.001). Between years 1 and 2, TL increased in the intervention arm (+76 bp) and decreased in the controls (-23 bp) (P=0.050). After two years, there was no difference between the arms (P=0.305).”

19) Unclear wording: does linear growth refer to telomeres or the child's height/length? (Abstract) If the latter, then this was not examined in this manuscript and should not go in the Abstract.

Linear growth refers to the child’s length/height, the primary outcomes of the main trial. We have deleted this sentence from the Abstract.